# REVISITING OUT-OF-DISTRIBUTION DETECTION: A SIMPLE BASELINE IS SURPRISINGLY EFFECTIVE

## ABSTRACT

It is an important problem in trustworthy machine learning to recognize out-of-distribution (OOD) inputs which are inputs unrelated to the in-distribution task. Many out-of-distribution detection methods have been suggested in recent years. The goal of this paper is to recognize common objectives as well as to identify the implicit scoring functions of different OOD detection methods. In particular, we show that binary discrimination between in- and (different) out-distributions is equivalent to several different formulations of the OOD detection problem. When trained in a shared fashion with a standard classifier, this binary discriminator reaches an OOD detection performance similar to that of Outlier Exposure. Moreover, we show that the confidence loss which is used by Outlier Exposure has an implicit scoring function which differs in a non-trivial fashion from the theoretically optimal scoring function in the case where training and test out-distribution are the same, but is similar to the one used when training with an extra background class. In practice, when trained in exactly the same way, all these methods perform similarly and reach state-of-the-art OOD detection performance.

## 1 INTRODUCTION

While deep learning has significantly improved performance in many application domains, there are serious concerns for using deep neural networks in applications which are of safety-critical nature. With one major problem being adversarial samples (Szegedy et al., 2014; Madry et al., 2018), which are small imperceptible modifications of the image that change the decision of the classifier, another major problem are overconfident predictions (Nguyen et al., 2015; Hendrycks & Gimpel, 2017; Hein et al., 2019) for images not belonging to the classes of the actual task. Here, one distinguishes between far out-of-distribution data, e.g. different forms of noise or completely unrelated tasks like CIFAR-10 vs. SVHN, and close out-of-distribution data which can for example occur in related image classification tasks where the semantic structure is very similar e.g. CIFAR-10 vs. CIFAR-100. Both are important to be distinguished from the in-distribution, but it is conceivable that close out-of-distribution data is the more difficult problem with potentially fatal consequences: in an automated diagnosis system we want that the system recognizes that it "does not know" when a new unseen disease comes in rather than assigning high confidence into a known class leading to fatal treatment decisions. Thus out-of-distribution awareness is a key property of trustworthy AI systems.

In this paper, we focus on the setting of OOD detection where during training time, there is no information available on the distribution of OOD inputs that might appear when the model is used for inference. A large number of different approaches to OOD detection based on combinations of density estimation, classifier confidence, logit space energy, feature space geometry, behaviour on auxiliary tasks, and other principles has been proposed to tackle this problem. We give a detailed overview of existing OOD detection methods in Appendix D. However, most OOD detection papers are focused on establishing superior empirical detection performance and provide little theoretical background on differences but also similarities to existing methods. In this paper we want to take a different path as we believe that a solid theoretical basis is needed to make further progress in this field. Our goal is to identify, at least for a particular subclass of techniques, whether the differences are indeed due to a different underlying theoretical principle or whether they are due to the efficiency of different *estimation techniques* for the same underlying detection criterion, called "scoring function". In some cases, we will see that one can even disentangle the estimation procedure from the scoring function, so that one can simulate several different scoring functions from one model's estimated quantities.

A simple approach to OOD detection is to treat it as a binary discrimination problem between in- and out-of-distribution, or more generally predicting a score how likely the input is OOD. In this paper, we show that from the perspective of Bayesian decision theory, several established methods are indeed equivalent to this binary discriminator. Differences arise mainly from i) the choice of the training out-distribution, e.g. the popular Outlier Exposure of Hendrycks et al. (2019a) has advocated the use of a rich and large set of natural images as a proxy for the distribution of natural images, and ii) differences in the estimation procedure. Concretely, the main contributions of this paper are:

- We show that several OOD detection approaches are equivalent to the binary discriminator between in- and out-distribution when analyzing the rankings induced by the Bayes optimal classifier/density.
- We derive the implicit scoring functions for the confidence loss (Lee et al., 2018a) used by Outlier Exposure (Hendrycks et al., 2019a) and for using an additional background class for the out-distribution (Thulasidasan et al., 2021). The confidence scoring function turns out not to be equivalent to the "optimal" scoring function of the binary discriminator when training and test out-distributions are the same.
- We show that when training the binary discriminator between in- and out-distribution together with a standard classifier on the in-distribution in a shared fashion, the binary discriminator reaches state-of-the-art OOD detection performance.
- We show that density estimation is equivalent to discrimination between the in-distribution and uniform noise which explains the frequent observation that standard density estimates are not suitable for OOD detection.

Even though we identify that a simple baseline is competitive with the state-of-the-art, the main aim of this paper is a better understanding of the key components of different OOD detection methods and to identify the key properties which lead to SOTA OOD detection performance. All of our findings are supported by extensive experiments on CIFAR-10 and CIFAR-100 with evaluation on various challenging out-of-distribution test datasets.

## 2 Models for OOD Data and Equivalence of OOD Detection Scores

As most work in the literature we consider OOD detection on a compact input domain $X$ where the most important example is image classification where $X = [0, 1]^D$. The most popular approach to OOD detection is the construction of an in-distribution-scoring function $f : X \to \mathbb{R} \cup \{\pm\infty\}$ such that $f(x)$ tends to be smaller if $x$ is drawn from an out-distribution than if it is drawn from the in-distribution. There is a variety of different performance metrics for this task, with a very common one being the *area under the receiver-operator characteristic curve* (AUC). The AUC for a scoring function $f$ distinguishing between an in-distribution $p(x|i)$ and an out-distribution $p(x|o)$ is given by

$$\text{AUC}_f\big(p(x|i), p(x|o)\big) = \mathop{\mathbb{E}}_{\substack{x \sim p(x|i) \\ y \sim p(y|o)}} \left[ \mathbb{1}_{f(x)>f(y)} + \frac{1}{2}\mathbb{1}_{f(x)=f(y)} \right]. \tag{1}$$

We define an equivalence of scoring functions based on their AUCs and will show that this equivalence implies equality of other employed performance metrics as well.

**Definition 1.** *Two scoring functions $f$ and $g$ are equivalent and we write $f \cong g$ if*

$$\text{AUC}_f\big(p(x|i), p(x|o)\big) = \text{AUC}_g\big(p(x|i), p(x|o)\big) \tag{2}$$

*for all potential distributions $p(x|i)$ and $p(x|o)$.*

As the AUC is not dependent on the actual values of $f$ but just on the ranking induced by $f$ one obtains the following characterization of the equivalence of two scoring functions.

**Theorem 1.** *Two scoring functions $f, g$ are equivalent $f \cong g$ if and only if there exists a strictly monotonously increasing function $\phi : \text{range}(g) \to \text{range}(f)$, such that $f = \phi(g)$.*

**Corollary 1.** *The equivalence between scoring functions in Def. 1 is an equivalence relation.*

Another metric is the *false positive rate at a fixed true positive rate q*, denoted as FPR@qTPR. A commonly used value for the TPR is 95%. The smaller the FPR@qTPR, the better the OOD discrimination performance.

**Lemma 1.** *Two equivalent scoring functions $f \cong g$ have the same FPR@qTPR for any pair of in- and out-distributions $p(x|i), p(x|o)$ and for any chosen TPR q.*

In the next section, we use the previous results to show that the Bayes optimal scoring functions of, several proposed methods for out-of-distribution detection are equivalent to the scoring functions of simple binary discriminators.

## 3 BAYES-OPTIMAL BEHAVIOUR OF BINARY DISCRIMINATORS AND COMMON OOD DETECTION METHODS

In the following we will show that the Bayes optimal function of several existing approaches to OOD detection for unlabeled data are equivalent to a binary discriminator between in- and a (training) out-distribution whereas differences arise when one has labeled data. As the equivalences are based on the Bayes optimal solution, these are asymptotic statements and thus it has to be noted that convergence to the Bayes optimal solution can be infinitely slow and that the methods can have implicit inductive biases. This is why we additionally support our findings with extensive experiments.

### 3.1 OOD DETECTION FOR METHODS USING UNLABELED DATA ONLY

We first provide a formal definition of OOD detection before we show the equivalence of density estimators resp. likelihood to a binary discriminator.

**The OOD problem** In order to make rigorous statements about the OOD detection problem we first have to provide the mathematical basis for doing so. We assume that we are given an in-distribution $p(x|i)$ and potentially also a *training* out-distribution $p(x|o)$. At this particular point no labeled data is involved, so both of them are just distributions over $X$. For simplicity we assume in the following that they both have a density wrt. the Lebesgue measure on $X = [0,1]^d$. We assume that in practice we get samples from the mixture distribution

$$p(x) = p(x|i)p(i) + p(x|o)p(o) = p(x|i)p(i) + p(x|o)(1 - p(i)) \tag{3}$$

where $p(i)$ is the probability that we expect to see in-distribution samples in total. In order to make the decision between in-and out-distribution for a given point $x$ it is then optimal to consider

$$p(i|x) = \frac{p(x|i)p(i)}{p(x)} = \frac{p(x|i)p(i)}{p(x|i)p(i) + p(x|o)p(o)}, \tag{4}$$

which is defined for all $x \in [0,1]^d$ with $p(x) > 0$ (assuming $p(x|i)$ and $p(x|o)$ can be written as densities). If the training out-distribution is also the test out-distribution then this is already optimal but we would like that the approach generalizes to other unseen test out-distributions and thus an important choice is the training out-distribution $p(x|o)$. Note that as $p(i|x)$ is only well-defined for all $x$ with $p(x) > 0$, it is thus reasonable to choose for $p(x|o)$ a distribution with support in $[0,1]^d$, that is $p(x|o) > 0$ for all $x \in [0,1]^d$. In this case we ensure that the criterion with which we perform OOD detection is defined for any possible input $x$. This is desirable as OOD detection should work for any possible input $x \in X$.

**Optimal prediction of a binary discriminator between in- and out-distribution** We consider a binary discriminator with model parameters $\theta$ between in- and (training) out-distribution, where $\hat{p}_\theta(i|x)$ is the predicted probability for the in-distribution. Under the assumption that $p(i)$ is the probability for in-distribution samples and using cross-entropy (which in this case is the logistic loss up to a constant global factor of $\log(2)$) the expected loss becomes:

$$\min_\theta p(i) \mathop{\mathbb{E}}_{x \sim p(x|i)} [-\log \hat{p}_\theta(i|x)] + p(o) \mathop{\mathbb{E}}_{x \sim p(x|o)} [-\log(1 - \hat{p}_\theta(i|x))] . \tag{5}$$

One can derive that the Bayes optimal classifier minimizing the expected loss has the predictive distribution:

$$\hat{p}_{\theta^*}(i|x) = \frac{p(x|i)p(i)}{p(x|i)p(i) + p(x|o)p(o)} = p(i|x). \tag{6}$$

Thus at least for the training out-distribution a binary classifier based on samples from in- and (training) out-distribution would suffice to solve the OOD detection problem perfectly.

**Equivalence of density estimation and binary discrimination for OOD detection**    In this section we further analyze the relationship of common OOD detection approaches with the binary discriminator between in-and out-distribution. We start with density estimators sourced from generative models. A basic approach that is known to yield relatively weak OOD performance (Nalisnick et al., 2019; Ren et al., 2019; Xiao et al., 2020) is directly utilizing a model's estimate for the density $p(x|i)$ at a sample input $x$. An improved density based approach which uses perturbed in-distribution samples as a surrogate training out-distribution is the Likelihood Ratios method (Ren et al., 2019), which proposes to fit a generative model for both the in- and out-distribution and to use the ratio between the likelihoods output by the two models as a discriminative feature.

We show that with respect to the scoring function, the correct density $p(x|i)$ is equivalent to the Bayes optimal prediction of a binary discriminator between the in-distribution and uniform noise. Furthermore, the density ratio $\frac{p(x|i)}{p(x|o)}$ is equivalent to the prediction of a binary discriminator between the two distributions on which the respective models used for density estimation have been trained. Because of this equivalence, we argue that the use of binary discriminators is a simple alternative to these methods because of its easier training procedure. While this equivalence is an asymptotic statement, the experimental comparisons in the appendix show that the methods perform similarly poorly compared to the methods using labeled data.

We first prove the more general case of arbitrary likelihood ratios. In the following we use the abbreviation $\lambda = \frac{p(o)}{p(i)}$ to save space and make the statements more concise.

**Lemma 2.** *Assume that $p(x|i)$ and $p(x|o)$ can be represented by densities and the support of $p(x|o)$ covers the whole input domain $X$. Then $\frac{p(x|i)}{p(x|o)} \cong \frac{p(x|i)}{p(x|i)+\lambda p(x|o)}$ for any $\lambda > 0$.*

This means that the likelihood ratio score of two optimal density estimators is equivalent to the in-distribution probability $\hat{p}_{\theta*}(i|x)$ predicted by a binary discriminator and this is true for any possible ratio of $p(i)$ to $p(o)$. In the experiments below, we show that using such a discriminator has similar performance as the likelihood ratios of the different trained generative models.

For the approaches that try to directly use the likelihood of a generative model as a discriminative feature, this means that their objective is equivalent to training a binary discriminator against uniform noise, whose density is $p_{\text{Uniform}}(x) = p(x|o) = 1$ at any $x$.

**Lemma 3.** *Assume that $p(x|i)$ can be represented by a density. Then $p(x|i) \cong \frac{p(x|i)}{p(x|i)+\lambda}$ for any $\lambda > 0$.*

This provides additional evidence why a purely density based approach for many applications proves to be insufficient as an OOD detection score on the complex image domain: it is not reasonable to assume that a binary discriminator between certain classes of natural images on the one hand and uniform noise on the other hand provides much useful information about images from other classes or even about other nonsensical inputs.

## 3.2   OOD DETECTION FOR METHODS USING LABELED DATA

We first discuss how one can formulate the OOD problem when one has access to labeled data for the in-distribution and we identify the target distribution of OOD detection using a background/reject class. Then we derive the Bayes optimal classifier of the confidence loss (Lee et al., 2018a) as used by the most successful variant of Outlier Exposure (Hendrycks et al., 2019a) and discuss the implicit scoring function. In most cases the scoring functions turn out not to be non-equivalent to $p(i|x)$ (which is optimal if training and test out-distribution agree) as they integrate additional information from the classification task.

**Bayes optimal solutions for OOD Detection with Background class and confidence loss Outlier Exposure**    Given a joint in-distribution $p(y, x|i)$ (where $y \in \{1, \ldots, K\}$ given that we have $K$ labels) for the labeled in-distribution, there are different ways how to come up with a joint distribution for in- and out-distribution. Interestingly, the different encodings used e.g. in training with a

background class (Thulasidasan et al., 2021) vs. training a classifier with confidence loss (Lee et al., 2018a) together with variants of the employed scoring function lead to methods which unexpectedly can have quite different behavior.

**Background class:** In this case we just put all out-of-distribution samples into a $K + 1$-class which is typically called background/reject class (Thulasidasan et al., 2021). The joint distribution then becomes

$$p(y, x) = \begin{cases} p(y, x|i)p(i) & \text{if } y \in \{1, \ldots, K\}, \\ p(x|o)p(o) & \text{if } y = K + 1. \end{cases}$$

We denote by $p(x|i) = \sum_{y=1}^{K} p(y, x|i)$ the marginal in-distribution and note that the marginal distribution of the joint distribution of in- and out-distribution is again given by

$$p(x) = p(x|i)p(i) + p(x|o)p(o).$$

Then we get the conditional distribution

$$p(y|x) = \begin{cases} p(y|x, i)p(i|x) & \text{if } y \in \{1, \ldots, K\}, \\ p(o|x) = 1 - p(i|x) & \text{if } y = K + 1. \end{cases}$$

The Bayes optimal solution of training with a background class using any calibrated loss function $L(y, f(x))$, e.g. the cross-entropy loss (Laptev et al., 2016), then yields a Bayes optimal classifier $f^*$ which has a predictive distribution $p_{f^*}(y|x) = p(y|x)$. There are two potential scoring function that come to mind:

$$s_1(x) = 1 - p_{f^*}(K + 1|x) \quad \text{and} \quad s_2(x) = \max_{k=1,\ldots,K} p_{f^*}(k|x)$$

The first one, used in Chen et al. (2021); Thulasidasan et al. (2021), is motivated by the fact that $p_{f^*}(K + 1|x)$ is directly the predicted probability that the point is from the out-distribution as indeed it holds: $s_1(x) = p(i|x)$ which is the optimal scoring function if training and test out-distribution are equal. On the other hand the maximal predicted probability $\max_{k=1,\ldots,K} p_{f^*}(k|x)$, which is often employed as a scoring function Hendrycks & Gimpel (2017), becomes for the Bayes optimal classifier

$$s_2(x) = p(i|x) \max_{k=1,\ldots,K} p(k|x, i),$$

which is a product of $p(i|x)$ and the maximal conditional probability of some class of the in-distribution (note that $s_2$ is well defined as $p(i|x)$ is defined if $p(x|o)$ has support everywhere in $X$ and if $p(i|x) > 0$ then also $p(x|i) > 0$). Thus the scoring function $s_2(x)$ integrates additionally to $p(i|x)$ also class-specific information and is thus less dependent on the chosen training out-distribution. In fact, one can see that $s_2$ only ranks points high if both the binary discriminator *and* the classifier rank the corresponding point high. On the other hand in the case where training and test out-distribution are identical, this scoring function is not equivalent to $p(i|x)$ and thus introduces a bias in the estimation.

**Outlier Exposure Hendrycks et al. (2019a) with confidence loss (Lee et al., 2018a):** we analyze the Bayes optimal solution for the confidence loss (Lee et al., 2018a) that is used by Outlier Exposure (OE) and show that the associated scoring function can be written, similarly to the scoring function $s_2(x)$ for training with a background class, as a function of $p(i|x)$ and $p(y|x, i)$.

The training objective with the confidence loss is in expectation given by

$$\min_{\theta} \mathbb{E}_{(x,y) \sim p(x,y|i)} \left[ \mathcal{L}_{\text{CE}}(f_\theta(x), y) \right] + \lambda \mathbb{E}_{x \sim p(x|o)} \left[ \mathcal{L}_{\text{CE}}(f_\theta(x), u^K) \right], \tag{7}$$

where $\theta$ are the model parameters and $f_\theta(x) \in \mathbb{R}^K$ is the model output as logits, and $u^K = (\frac{1}{K}, \ldots, \frac{1}{K})^T$ is the uniform distribution over the $K$ classes of the in-distribution classification task.

In the following theorem we derive the Bayes optimal predictive distribution for this training objective.

**Theorem 2.** *The predictive distribution $p_{f^*}(y|x)$ of the Bayes optimal classifier $f^*$ minimizing the expected confidence loss is given for $y \in \{1, \ldots, K\}$ as*

$$p_{f^*}(y|x) = p(i|x)p(y|x, i) + \frac{1}{K}\big(1 - p(i|x)\big). \tag{8}$$

Thus the effective scoring function of using the probability of the predicted class as suggested in Hendrycks & Gimpel (2017); Lee et al. (2018a); Hendrycks et al. (2019a) is given by

$$s_3(x) = p(i|x) \max_{y=1,\ldots,K} p(y|x,i) + \frac{1}{K}\big(1 - p(i|x)\big) = p(i|x)\Big[\max_{y=1,\ldots,K} p(y|x,i) - \frac{1}{K}\Big] + \frac{1}{K}.$$

Please note that the term inside the brackets is positive as $\max_{k=1,\ldots,K} p(k|x,i) \geq \frac{1}{K}$. Interestingly, the scoring functions $s_2$ and $s_3$ are not equivalent even though they look quite similar. In particular, due to the subtraction of $\frac{1}{K}$ the scoring function $s_3$ puts more emphasis on the classifier than $s_2$. In Appendix F we additionally analyze Energy Based OOD Detection (Liu et al., 2020), and show that the Bayes optimal decision is equivalent to using the scoring function $s_1 = p(i|x)$.

### 3.3 SEPARATE VERSUS SHARED ESTIMATION OF $p(i|x)$ AND $p(y|x,i)$

So far we have derived that at least from the point of view of the ranking induced by the Bayes optimal solution, OOD detection based on generative methods, likelihood ratios and the background class formulation with the scoring function $s_1$ is equivalent to a binary classification problem between in- and out-distribution in order to estimate $p(i|x)$. The differences arise mainly in the choice of the training out-distribution $p(x|o)$: i) uniform for generative resp. density based methods, ii) a quite specific out-distribution for likelihood ratios (Ren et al., 2019) and iii) a proxy of the distribution of all natural images (Hendrycks et al., 2019a; Thulasidasan et al., 2021).

On the other hand when labeled data is involved we can additionally train a classifier on the in-distribution in order to estimate $p(y|x,i)$. We will then combine the estimates of $p(i|x)$ and $p(y|x,i)$ according to the three scoring functions derived in the previous section and check if the novel OOD detection methods constructed in this way perform similar to the OOD methods from which we derived the corresponding scoring function i) OOD detection with a background class (Thulasidasan et al., 2021) or ii) using Outlier Exposure Hendrycks et al. (2019a). This will allow us to differentiate between differences of the employed scoring functions for OOD detection and the estimators for the involved quantities. In this way we foster a more systematic approach to OOD detection.

In the unlabeled case we train simply the binary classifier $p_\theta : [0,1]^d \to \mathbb{R}$ using logistic/cross entropy loss in a class balanced fashion

$$\min_\theta -\frac{1}{N}\sum_{i=1}^N \log\big(\hat{p}_\theta(i|x_i^{\text{IN}})\big) - \frac{\lambda}{M}\sum_{j=1}^M \log\big(1 - \hat{p}_\theta(i|x_j^{\text{OUT}})\big) \ , \tag{9}$$

where $(x_i^{\text{IN}})_{i=1}^N$ and $(x_j^{\text{OUT}})_{j=1}^M$ are samples from the in-distribution and the out-distribution.

In the case where we have labeled data we can additionally solve the classification problem. The obvious approach is to train the binary classifier for estimating $p(i|x)$ and the classifier to estimate $p(y|x,i)$ completely independently. Not surprisingly, we show in Section 4 that this approach works less well. In fact both tasks benefit from each other. Moreover, in training a neural network using a background class or with Outlier Exposure (Hendrycks et al., 2019a) we are implicitly using a shared representation for both tasks which improves the results.

Thus we propose to train the binary discriminator of in-versus out-distribution together with the classifier on the in-distribution jointly. Concretely, we use a neural network with $K+1$ outputs where the first $K$ outputs represent the classifier and the last output is the logit of the binary discriminator. The resulting shared problem can then be written as

$$\min_\theta -\frac{1}{N_b}\sum_{r=1}^{N_b} \log\big(\hat{p}_\theta(i|x_r^{\text{IN}})\big) - \frac{\lambda}{M}\sum_{s=1}^M \log\big(1 - \hat{p}_\theta(i|x_s^{\text{OUT}})\big) - \frac{1}{N_c}\sum_{t=1}^{N_c} \log\big(\hat{p}_\theta(y_t^{\text{IN}}|x_t^{\text{IN}})\big) \ , \tag{10}$$

where $\lambda = \frac{p(o)}{p(i)}$ which is typically set to 1 during training in order to get a class-balanced problem. Note that the in-distribution samples $(x_r^{\text{IN}})_{r=1}^{N_b}$ used to estimate $p(i|x)$ can be a super-set of the labeled examples $(x_t^{\text{IN}}, y_t^{\text{IN}})_{t=1}^{N_c}$ used to train the classifier so that one can potentially integrate unlabeled data - this is an advantage compared to OOD detection with a background class or Outlier Exposure where this is not directly possible. We stress that the loss functions of the classifier and the discriminator act on independent outputs; the functions modelling the two tasks only interact with each other due to the shared network weights up to the final layer. Nevertheless, we see in the next Section 4 that training with a shared representation boosts both the classifier and the binary discriminator.

Table 1: Accuracy on the in-distribution (CIFAR-10/CIFAR-100) and FPR@95%TPR for various test out-distributions of different OOD methods with OpenImages as training out-distribution (results for the test set of OpenImages are not used in the mean FPR). Lower false positive rate is better. All methods except Mahalanobis have been trained using the same architecture, training parameters, schedule and augmentation. $s_1, s_2, s_3$ are the scoring functions introduced in Section 3.2. Our binary discriminator (BINDISC) resp. the combination with the shared classifier (SHARED COMBI) and the models with background class (BGC) with scoring functions $s_2$ or $s_3$ outperform the Mahalanobis detector (Lee et al., 2018b) and are similar to Outlier Exposure (Hendrycks et al., 2019a). CelebA makes no sense as test out-distribution for CIFAR-100 as man/woman are classes in CIFAR-100.

IN-DISTRIBUTION: CIFAR-10

| MODEL | ACC. | MEAN FPR | SVHN FPR | LSUN FPR | UNI FPR | SMOOTH FPR | C-100 FPR | 80M FPR | CELA FPR | OPENIM FPR |
|---|---|---|---|---|---|---|---|---|---|---|
| PLAIN CLASSI | 95.16 | 53.01 | 47.87 | 50.00 | 17.51 | 65.81 | 60.43 | 53.44 | 76.00 | 63.71 |
| MAHALANOBIS | | 36.68 | 20.97 | 49.00 | 0.00 | 0.00 | 57.21 | 48.85 | 80.71 | 55.13 |
| OE | 95.06 | **15.20** | 9.58 | 0.00 | 0.00 | 0.00 | 54.05 | 42.33 | 0.45 | 3.46 |
| BGC $s_1$ | | 18.83 | 2.36 | 0.00 | 0.00 | 0.00 | 72.00 | 56.41 | 1.04 | 0.05 |
| BGC $s_2$ | 95.21 | 16.52 | 7.51 | 0.00 | 0.05 | 2.10 | 55.16 | 44.57 | 6.26 | 1.65 |
| BGC $s_3$ | 95.21 | 16.63 | 7.69 | 0.00 | 0.07 | 2.36 | 55.19 | 44.67 | 6.41 | 1.74 |
| SHARED BINDISC | | 19.56 | 4.65 | 0.00 | 0.00 | 0.00 | 77.50 | 53.93 | 0.87 | 0.04 |
| SHARED CLASSI | **95.28** | 29.34 | 28.00 | 7.00 | 2.33 | 33.04 | 58.61 | 47.90 | 28.54 | 35.94 |
| SHARED COMBI $s_2$ | **95.28** | 16.00 | 8.56 | 0.00 | 0.00 | 0.00 | 58.80 | 42.79 | 1.83 | 0.61 |
| SHARED COMBI $s_3$ | **95.28** | 16.06 | 9.00 | 0.00 | 0.00 | 0.00 | 58.68 | 42.85 | 1.91 | 0.66 |

IN-DISTRIBUTION: CIFAR-100

| MODEL | ACC. | MEAN FPR | SVHN FPR | LSUN FPR | UNI FPR | SMOOTH FPR | C-10 FPR | 80M FPR | OPENIM FPR |
|---|---|---|---|---|---|---|---|---|---|
| PLAIN CLASSI | 77.16 | 67.57 | 75.50 | 78.33 | 22.60 | 70.98 | 80.55 | 77.43 | 80.80 |
| MAHALANOBIS | | 53.88 | 54.36 | 66.00 | 46.43 | 0.06 | 85.39 | 71.01 | 74.69 |
| OE | 77.19 | 35.03 | 47.36 | 0.00 | 0.67 | 0.08 | 84.64 | 77.42 | 1.28 |
| BGC $s_1$ | | **31.14** | 11.58 | 0.00 | 0.00 | 0.00 | 93.94 | 81.29 | 0.07 |
| BGC $s_2$ | **77.61** | 33.32 | 37.06 | 0.00 | 0.00 | 0.20 | 84.50 | 78.17 | 1.26 |
| BGC $s_3$ | **77.61** | 33.36 | 37.27 | 0.00 | 0.00 | 0.20 | 84.51 | 78.19 | 1.27 |
| SHARED BINDISC | | 31.86 | 10.77 | 0.00 | 0.00 | 0.00 | 95.25 | 85.11 | 0.08 |
| SHARED CLASSI | 77.35 | 67.23 | 71.05 | 5.00 | 97.70 | 69.68 | 82.05 | 77.89 | 28.38 |
| SHARED COMBI $s_2$ | 77.35 | 33.01 | 37.30 | 0.00 | 0.00 | 1.06 | 82.71 | 77.01 | 1.80 |
| SHARED COMBI $s_3$ | 77.35 | 33.06 | 37.57 | 0.00 | 0.00 | 1.13 | 82.68 | 77.01 | 1.85 |

## 4 EXPERIMENTS

We use CIFAR-10 and CIFAR-100 (Krizhevsky & Hinton, 2009) datasets as in-distribution and OpenImages dataset (Krasin et al., 2017) as training out-distribution. The 80 Million Tiny Images (80M) dataset (Torralba et al., 2008) is the de facto standard for training out-distribution aware models that has been adopted by most prior works, but this dataset has been withdrawn by the authors as Birhane & Prabhu (2021) pointed out the presence of offensive images. To be able to compare with other state-of-the-art methods without introducing a potential bias due to dataset selection, we include the evaluation with 80M as training out-distribution in Appendix H. Moreover, we show in the appendix results for the binary discriminator trained with different training out-distributions vs. likelihoods resp. likelihood ratios (Ren et al., 2019) as OOD method.

We use as OOD detection metric the false positive rate at 95% true positive rate, FPR@95%TPR; evaluations with AUC are in Appendix G. We evaluate the OOD detection performance on the following datasets: SVHN (Netzer et al., 2011), resized LSUN Classroom (Yu et al., 2015), Uniform Noise, Smooth Noise generated as described by (Hein et al., 2019), the respective other CIFAR dataset, 80M, and CelebA (Liu et al., 2015). We highlight that none of the listed methods has access

to those test distributions during training or for fine-tuning as we try to assess the ability of an out-distribution aware model to generalize to unseen distributions. The FPR for the OpenImages test set is not included in the Mean AUC, since this distribution *has* been used during training.

The binary discriminators (BINDISC) as well as the classifiers with background class (BGC) and the shared binary discriminator+classifier (SHARED) of $p(i|x)$ and $p(y|x, i)$ are trained on the 40-2 Wide Residual Network (Zagoruyko & Komodakis, 2016) architecture with the same training schedule as used in Hendrycks et al. (2019a) for training their Outlier Exposure(OE) models. This includes averaging the loss over batches that are twice as large for the out-distribution. This way we ensure that the differences do not arise due to differences in the training schedules or other important details but only on the employed objectives. In addition to their standard augmentation and normalization, we apply AutoAugment (Cubuk et al., 2019) without Cutout, and we use $\lambda = 1$ where applicable, which is a sound choice as we observe in an ablation on $\lambda$ in Appendix K. For the Mahalanobis OOD detector (Lee et al., 2018b), we use the models and code published by the authors and use OpenImages for the fine tuning of input noise and layer weighting regression. We describe the exact details of the training settings and the used dataset splits in Appendix C.

### 4.1 OUT-DISTRIBUTION AWARE TRAINING WITH LABELED IN-DISTRIBUTION DATA

In Table 1 we compare multiple OOD methods trained with training out-distribution OpenImages and CIFAR-10/100 as in-distribution: confidence of standard training (PLAIN) and OE, MAHALANOBIS detection, classifier with background class (BGC) and the combination of a plain classifier and a binary in-vs-out-distribution classifier with shared representation (SHARED COMBI). As described in Section 2, both BGC and SHARED COMBI can be used in combination with different scoring functions. For BGC, we evaluate all three scoring functions $s_1$, $s_2$ and $s_3$ and for SHARED COMBI we only use $s_2$ and $s_3$ as $s_1$ is equivalent to $p(i|x)$ which is the output of SHARED BINDISC. Additionally, we evaluate OOD detection based on the confidence of the shared classifier (SHARED CLASSI) trained together with SHARED BINDISC.

For CIFAR-10, a first interesting observation is that SHARED CLASSI has remarkably good OOD performance; significantly better than a normal classifier (plain) even though it is just trained using normal cross-entropy loss and so the OOD performance is only due to the regularization enforced by the shared representation with SHARED BINDISC. In fact SHARED BINDISC has already good OOD performance with a mean FPR@95%TPR of 19.56, which is improved by considering scoring function $s_2/s_3$ in the combination of SHARED BINDISC and SHARED CLASSI which yields very good classification accuracy and mean FPR/AUC. Moreover, interesting are the results of the classifier with background class (BGC) which is the method recently advocated in Thulasidasan et al. (2021). It works very well but the performance depends on the chosen scoring function. Whereas $s_1$ (output of the background class) is a usable scoring function (mean FPR: 18.83), the maximum probability over the other classes $s_2$ (mean FPR: 16.52) or the combination in terms of $s_3$ (mean FPR: 16.63) performs better. In total with the scoring function $s_2/s_3$ integrating classifier and discriminative information, BGC reaches similar performance to OE (which implicitly also uses $s_3$ as scoring function). In general, the differences of the methods are relatively minor both in terms of OOD detection and classification accuracy, where the latter is better for all OOD methods compared to the plain classifier; this is most likely explained by better learned representations, see also Hendrycks et al. (2019a); Augustin et al. (2020) for similar observations. The results for CIFAR-100 are similar to CIFAR-10, with some reversals of the overall rankings of the compared methods. OE achieves comparable OOD results to BGC $s_2/s_3$ and SHARED COMBI $s_2/s_3$. For this in-distribution our BGC $s_1$ and SHARED BINDISC perform best in terms of OOD performance. Classification test accuracy is slightly higher for BGC and SHARED, but the differences are minor. The experiments with 80M as training out-distribution (Table 7 in Appendix H) confirm these observations.

Overall, as suggested by the theoretical results on the equivalence of the Bayes optimal classifier of OE with the $s_3$ scoring function of BGC and SHARED COMBI, we observe that even though these methods are derived and in particular trained with quite different objectives, they behave very similar in our experiments. In total we think that this provides a much better understanding where differences of OOD methods are coming from. Regarding the question of which method and scoring function should be used for a given application, the experimental results across datasets and different out-distributions, see Appendix H, suggest that their difference is minor and there is no clear best choice. However, in Appendix B, we describe a potential situation where the $s_3$ score and in

consequence OE is not powerful enough to distinguish in- and out-of-distribution inputs. On the other hand, in cases where the $s_1$ score is not very informative as training and test out-distributions largely differ, combining it with the classifier confidences is beneficial; this can be observed in experiments with SVHN as training out-distribution which we show in Appendix I. This is why for an unknown situation, we recommend BGC or SHARED COMBI with the $s_2$ scoring function as the safest option. However, it is an open question if there are also situations where $s_2$ is fundamentally inferior to $s_3$.

## 4.2 SHARED REPRESENTATION LEARNING FOR THE BINARY DISCRIMINATOR

As highlighted above the shared training of SHARED CLASSI and SHARED BINDISC and their combination SHARED COMBI with $s_2/s_3$ as scoring functions yields strong OOD detection and test accuracy among all methods. Here, we evaluate the importance of training the binary discriminator and the plain classifier with a shared representation in comparison to training two entirely separate models PLAIN CLASSI and SEPARATE BINDISC and their combination SEPARATE COMBI with scoring function $s_3$. The results for CIFAR-10 and CIFAR-100 can be found in Table 2. In total, we see that separate training in particular for CIFAR-100 leads to worse results compared to shared training as expected as the binary discriminator and the classifier cannot benefit from each other. An interesting curiosity is that the combination of the separate classifier with the binary discriminator trained in a shared fashion (PLAIN $\otimes$ SHA DISC) yields almost the same OOD results as SHARED COMBI even though the classifier is significantly worse. Overall, SHARED COMBI performs significantly better when also considering the better classification accuracy which it inherits from SHARED CLASSI.

Table 2: Evaluation (same metrics as in Table 1) of models trained with shared and separate representations. Shared training benefits both the classifier and the binary discriminators.

IN-DISTRIBUTION: CIFAR-10

| MODEL | ACC. | MEAN FPR | SVHN FPR | LSUN FPR | UNI FPR | SMOOTH FPR | C-100 FPR | 80M FPR | CELA FPR | OPENIM FPR |
|---|---|---|---|---|---|---|---|---|---|---|
| SEPARATE BINDISC | | 23.49 | 6.21 | 0.00 | 0.00 | 0.00 | 83.79 | 65.77 | 8.68 | 0.00 |
| PLAIN CLASSI | 95.16 | 53.01 | 47.87 | 50.00 | 17.51 | 65.81 | 60.43 | 53.44 | 76.00 | 63.71 |
| SEPARATE COMBI $s_3$ | 95.16 | 21.40 | 13.15 | 0.00 | 0.00 | 0.00 | 59.96 | 49.78 | 26.93 | 0.45 |
| SHARED BINDISC | | 19.56 | 4.65 | 0.00 | 0.00 | 0.00 | 77.50 | 53.93 | 0.87 | 0.04 |
| SHARED CLASSI | **95.28** | 29.34 | 28.00 | 7.00 | 2.33 | 33.04 | 58.61 | 47.90 | 28.54 | 35.94 |
| SHARED COMBI $s_3$ | **95.28** | 16.06 | 9.00 | 0.00 | 0.00 | 0.00 | 58.68 | 42.85 | 1.91 | 0.66 |
| PLAIN $\otimes$ SHA DISC $s_3$ | 95.16 | **15.96** | 8.10 | 0.00 | 0.00 | 0.00 | 58.48 | 42.60 | 2.53 | 0.70 |

IN-DISTRIBUTION: CIFAR-100

| MODEL | ACC. | MEAN FPR | SVHN FPR | LSUN FPR | UNI FPR | SMOOTH FPR | C-10 FPR | 80M FPR | OPENIM FPR |
|---|---|---|---|---|---|---|---|---|---|
| SEPARATE BINDISC | | 32.50 | 14.28 | 0.00 | 0.00 | 0.00 | 96.50 | 84.22 | 0.02 |
| PLAIN CLASSI | 77.16 | 67.57 | 75.50 | 78.33 | 22.60 | 70.98 | 80.55 | 77.43 | 80.80 |
| SEPARATE COMBI $s_3$ | 77.16 | 41.94 | 69.44 | 0.00 | 0.00 | 24.39 | 81.15 | 76.67 | 0.89 |
| SHARED BINDISC | | **31.86** | 10.77 | 0.00 | 0.00 | 0.00 | 95.25 | 85.11 | 0.08 |
| SHARED CLASSI | **77.35** | 67.23 | 71.05 | 5.00 | 97.70 | 69.68 | 82.05 | 77.89 | 28.38 |
| SHARED COMBI $s_3$ | **77.35** | 33.06 | 37.57 | 0.00 | 0.00 | 1.13 | 82.68 | 77.01 | 1.85 |
| PLAIN $\otimes$ SHA DISC $s_3$ | 77.16 | 33.38 | 37.00 | 0.00 | 0.00 | 5.42 | 81.33 | 76.50 | 2.23 |

## 5 CONCLUSION

In this paper we have analyzed different OOD detection methods and have shown that the simple baseline of a binary discriminator between in-and out-distribution is a powerful OOD detection method if trained in a shared fashion with a classifier. Moreover, we have revealed the inner mechanism of Outlier Exposure and training with a background class which unexpectedly use a scoring function which integrates information from $p(i|x)$ *and* $p(y|x,i)$. We think that these findings will allow to build novel OOD methods in a more principled fashion.

## 6 ETHICS AND REPRODUCIBILITY STATEMENT

In this paper we provide an explanation for the inner workings of established OOD detection methods and propose a novel OOD detection method based on these considerations which outperforms OE in terms of OOD performance and test accuracy. The final goal is to have more trustworthy classifiers. One could criticize that the focus on OOD FPR/AUC performance and test accuracy covers just certain aspects and other aspects like calibration of the classifiers, fairness, robustness to corruptions or adversarial attacks play an important role, too. However, apart from the usual dual use problem we see only positive societal aspects of our paper, as it leads to more trustworthy ML methods.

We discuss the problematic situation with the retracted 80 Million Tiny Images (Torralba et al., 2008) dataset – which is used by many previous works in the field – in Appendix H and replace the it with OpenImages (Krasin et al., 2017) for training the models in the main paper. For comparability with previous methods, we include evaluations of models trained on 80M in the appendix. We hope that introducing an alternative dataset for natural surrogate OOD training helps the community towards avoiding the retracted dataset in the future, both from an ethical and also a practical perspective, in case 80M becomes fully unavailable.

All experimental details including used hardware are given in Appendix C, and code for training and evaluating out methods as well as weights of the evaluated neural networks are available at `https://anonymous.4open.science/r/OOD_BGC_BinDisc-D7FB`.

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

## A  PROOFS

**Theorem 1.** *Two scoring functions $f, g$ are equivalent $f \cong g$ if and only if there exists a strictly monotonously increasing function $\phi : \mathrm{range}(g) \to \mathrm{range}(f)$, such that $f = \phi(g)$.*

*Proof.* Assume that such a function $\phi$ exists. Then for any pair $x, y$ we have the logical equivalences $g(x) > g(y) \Leftrightarrow f(x) = \phi(g(x)) > \phi(g(y)) = f(y)$ and $g(x) = g(y) \Leftrightarrow f(x) = \phi(g(x)) = \phi(g(y)) = f(y)$. This directly implies that the AUCs are the same, regardless of the distributions.

Assume $f \cong g$. For each $a \in \mathrm{range}(g)$, choose some $\hat{a} \in g^{-1}(a)$. For any pair $x, y \in X$, by regarding the Dirac distributions $p(x|i) = \delta_x$ and $p(x|o) = \delta_y$ that are each concentrated on one of the points, we can infer that $f(x) > f(y) \Leftrightarrow \mathrm{AUC}_f(p(x|i), p(x|o)) = 1 \Leftrightarrow \mathrm{AUC}_g(p(x|i), p(x|o)) = 1 \Leftrightarrow g(x) > g(y)$ and similarly $f(x) = f(y) \Leftrightarrow g(x) = g(y)$. The latter ensures that the function defined as

$$\phi : \mathrm{range}(g) \to \mathrm{range}(f)$$
$$a \mapsto f(\hat{a}) \tag{11}$$

is independent of the choice of $\hat{a}$ and that $f = \phi \circ g$, and the former confirms that $\phi$ is strictly monotonously increasing. $\qquad\square$

**Lemma 1.** *Two equivalent scoring functions $f \cong g$ have the same FPR@qTPR for any pair of in- and out-distributions $p(x|i), p(x|o)$ and for any chosen TPR q.*

*Proof.* We know that a function $\phi$ as in Theorem 1 exists. Then for any pair $x, y$, we have the logical equivalences

$$g(x) > g(y) \Leftrightarrow f(x) = \phi(g(x)) > \phi(g(y)) = f(y) \tag{12}$$

and

$$g(x) = g(y) \Leftrightarrow f(x) = \phi(g(x)) = \phi(g(y)) = f(y) . \tag{13}$$

This directly implies that the FPR@qTPR-values are the same, for any $p(x|i), p(x|o)$ and q. $\qquad\square$

**Lemma 2.** *Assume that $p(x|i)$ and $p(x|o)$ can be represented by densities and the support of $p(x|o)$ covers the whole input domain $X$. Then $\frac{p(x|i)}{p(x|o)} \cong \frac{p(x|i)}{p(x|i) + \lambda p(x|o)}$ for any $\lambda > 0$.*

*Proof.* The function $\phi : [0, \infty] \to [0, 1]$ defined by $\phi(x) = \frac{x}{x + \lambda}$ (setting $\phi(\infty) = 1$) fulfills the criterion from Theorem 1 of being strictly monotonously increasing. With

$$\phi\left(\frac{p(x|i)}{p(x|o)}\right) = \frac{\frac{p(x|i)}{p(x|o)}}{\frac{p(x|i)}{p(x|o)} + \lambda \frac{p(x|o)}{p(x|o)}} = \frac{p(x|i)}{p(x|i) + \lambda p(x|o)} \tag{14}$$

for $p(x|o) \neq 0$ and $\phi\left(\frac{p(x|i)}{0}\right) = \phi(\infty) = 1 = \frac{p(x|i)}{p(x|i) + \lambda \cdot 0}$, the equivalence follows. $\qquad\square$

**Lemma 3.** *Assume that $p(x|i)$ can be represented by a density. Then $p(x|i) \cong \frac{p(x|i)}{p(x|i) + \lambda}$ for any $\lambda > 0$.*

*Proof.* This is a special case of Lemma 2, by setting $p(x|o) = 1 = p_{\mathrm{Uniform}}(x)$. $\qquad\square$

**Theorem 2.** *The predictive distribution $p_{f^*}(y|x)$ of the Bayes optimal classifier $f^*$ minimizing the expected confidence loss is given for $y \in \{1, \ldots, K\}$ as*

$$p_{f^*}(y|x) = p(i|x)p(y|x, i) + \frac{1}{K}\left(1 - p(i|x)\right) . \tag{8}$$

*Proof.* Minimizing the loss of Outlier Exposure given in Eq. (7) means solving the optimization problem

$$\min_{p_f(\cdot|x)} \; -p(i|x) \cdot \sum_{k=1}^{K} p(k|x,i) \cdot \log p_f(k|x) - (1-p(i|x)) \cdot \sum_{k=1}^{K} \frac{1}{K} \cdot \log p_f(k|x)$$

$$\text{subject to} \quad p_f(k|x) \geq 0 \;\; \text{for each } k \in \{1, \ldots, K\} \tag{15}$$

$$\sum_{k=1}^{K} p_f(k|x) = 1 \,,$$

where $p_f(\cdot|x)$ is the model's $K$-dimensional prediction. For $p(i|x) = 0$ or $p(k|x,i) = 0$, the optimalities of the respective terms are easy to show (applying the common conventions for $0 \log 0$), so we assume that tose are non-zero. The Lagrange function of the optimization problem is

$$L(p_f(\cdot|x), \alpha, \beta) = -p(i|x) \cdot \sum_{k=1}^{K} p(k|x,i) \cdot \log p_f(k|x) - (1-p(i|x)) \cdot \sum_{k=1}^{K} \frac{1}{K} \cdot \log p_f(k|x)$$

$$- \sum_{k=1}^{K} \alpha_k p_f(k|x) + \beta \left( -1 + \sum_{k=1}^{K} p_f(k|x) \right) \,, \tag{16}$$

with $\beta \in \mathbb{R}$ and $\alpha \in \mathbb{R}_+^K$. Its first derivative with respect to $p_f(k|x)$ for any $k$ is

$$\frac{\partial L}{p_f(k|x)} = -p(i|x) \cdot p(k|x,i) \frac{1}{p_f(k|x)} - (1-p(i|x)) \cdot \frac{1}{K} \frac{1}{p_f(k|x)} - \alpha_k + \beta$$

$$= -\frac{s^K(k|x)}{p_f(k|x)} - \alpha_k + \beta \,, \tag{17}$$

where we set $s^K(k|x) := p(i|x)p(k|x,i) + \frac{1}{K}\big(1 - p(i|x)\big)$. The second derivative is a positive diagonal matrix for any point of its domain, therefore we find the unique minimum by setting (17) to zero, i.e. at

$$p_f(k|x) = \frac{s^K(k|x)}{\beta - \alpha_k} \,. \tag{18}$$

The dual problem is hence the maximization (with $\alpha_k \geq 0$) of

$$q(\alpha, \beta) = -p(i|x) \cdot \sum_{k=1}^{K} p(k|x,i) \cdot \log \frac{s^K(k|x)}{\beta - \alpha_k} - (1-p(i|x)) \cdot \sum_{k=1}^{K} \frac{1}{K} \cdot \log \frac{s^K(k|x)}{\beta - \alpha_k}$$

$$- \sum_{k=1}^{K} \alpha_k \frac{s^K(k|x)}{\beta - \alpha_k} + \beta \left( -1 + \sum_{k=1}^{K} \frac{s^K(k|x)}{\beta - \alpha_k} \right)$$

$$= \sum_{k=1}^{K} s^K(k|x) \left( -\log s^K(k|x) + \log(\beta - \alpha_k) + \frac{\beta}{\beta - \alpha_k} - \frac{\alpha_k}{\beta - \alpha_k} \right) - \beta \,;$$

here, $\alpha$ only appears in $\log(\beta - \alpha_k)$ which has a positive factor $s^K(k|x)$, so $\alpha = 0$ maximizes the expression. Noting $\sum_{k=1}^{K} s^K(k|x) = 1$, what remains is $q^0(\beta) = 1 + \log(\beta) - \sum_{k=1}^{K} s^K(k|x) \log s^K(k|x) - \beta$, which is maximized by $\beta = 1$. This means that the dual optimal pair is $p_f(k|x) = s^K(k|x), (\beta = 1, \alpha = 0)$. Slater's condition (Boyd et al., 2004) holds since the feasible set of the original problem is the probability simplex. Thus, $p_{f*}(\cdot|x) = s^K(x)$ is indeed primal optimal. □

## B  CONFIDENCE LOSS MODELS AND MODELS WITH A BACKGROUND CLASS OR A BINARY DISCRIMINATOR ARE ONLY EQUIVALENT AFTER $s_3$ IS APPLIED

Seeing that for both in-distribution accuracy and OOD detection, OE models trained with confidence loss, models with background class and shared classifier/discriminator combinations behave very

similarly, the question arises if the training methods themselves lead to equivalent models. One idea might be that and effect of the confidence loss on the degree of freedom from logit translation invariance could be unfolded to obtain a $K+1$-dimensional output that contains the same information as a classifier with background class or with and additional binary discriminator output (the latter two are indeed equivalent). This is not the case, as the following example that in certain situations, the $s_1$ and $s_2$ scores of background class models and classifier/discriminator combinations are able to separate in- and out-distribution, while the $s_3$ score and the equivalent confidence loss/OE models cannot.

## B.1 EURO COIN CLASSIFIER

As an example where the mentioned non-equivalence would occur, we hypothetically regard the task of classifying photos of 1-Euro coins by the issuing country. Each €1 coin features a common side that is the same for each country and a national side that pictures a unique motive per country. We assume that one side is visible on each photo, and that the training dataset of size $2mK$ is balanced, consisting of $2m$ coin photos with label $c$ for each country $c \in \{1, \ldots, K\}$, where $m$ photos show the common side and the other $m$ photos show the informative national side for each country $c$.

It is easy to see that the Bayes optimal classifier trained with cross-entropy loss on this dataset predicts $100\%$ for the respective country $c$ when shown a photo of the national side of a €1 coin, and predicts $1/K$ for each country when shown the common side of a €1 coin.

Now we compare the behaviour of the different methods given a training out-distribution of poker chips images which are clearly recognizable as not being €1 coins.

A $K$-class model trained with confidence loss (Lee et al., 2018a) will not make a difference between common side coin images and poker chip images, and in the Bayes optimal case, it will predict the uniform class distribution in both cases. This does not only hold for the prediction of a hypothetical Bayes optimal model: assuming full batch gradient descent and identical sets of $m$ common side training photos for each class, the loss for a common side input is the same as the loss for a poker chip.

On the other hand, a binary discriminator will easily distinguish between poker chips and €1 coins, no matter which side of the coin is shown. The same holds for a model with background class: the score of the class $K+1$ will be close to 1 for chips and close to 0 for €1 coins.

We conclude that in the described situation, models trained with confidence loss/outlier exposure are not able to sufficiently distinguish in- and out-distribution, while the $s_1$ scoring function of a classifier with background class or a binary discriminator is suitable for this task.

With the $s_2$ scoring function, the background class model gives us $s_2(x) = \max_{k=1,\ldots,K} p_f(k|x)$, and thus $\frac{(1-p_f(K+1|x))}{K} \le s_2(x) \le 1 - p_f(K+1|x)$, which means that if $p_f(K+1|x)$ is sufficiently large for in-distribution inputs and sufficiently small for out-of-distribution inputs, $s_2$ is able to distinguish them independent of inconclusiveness in the first $K$ classes. Similarly, $s_2$ applied to a binary discriminator with a classifier (shared trained or not) will be able to distinguish common sides of coins and poker chips.

With $s_3$, on the other hand, common sides of coins and poker chips can no longer be separated. For a classifier/discriminator pair, as defined above, $s_3(x) = p_d(i|x)\left[\max_{y=1,\ldots,K} p_c(y|x,i) - \frac{1}{K}\right] + \frac{1}{K}$. If on the common side of a coin the classifier predicts uniform $\frac{1}{K}$, we have $s_3(x) = \frac{1}{K}$ no matter what the discriminator $p_d(i|x)$ predicts. On poker chips with discriminator prediction $p_d(i|x) = 1$, we also get $s_3(x) = 1$. For background class models, $s_3(x) = \max_{k=1,\ldots,K} p_f(k|x) + \frac{1}{K}p_f(K+1|x)$ also yields $\frac{1}{K}$ for a common side where the prediction over the $K$ in-distribution classes is uniform and for a poker chips, where $p_f(K+1|x) = 1$. The fact that in this coin scenario when scored with $s_3$, background class and classifier/discriminator combinations have the same problem as confidence loss/OE is not surprising considering their equivalence shown in Theorem 2.

## C  EXPERIMENTAL DETAILS

### C.1  TRAINING

For training our models, we build upon the code of Hendrycks et al. (2019a) which they have available at `https://github.com/hendrycks/outlier-exposure` and borrow their general architecture and training settings. Concretely, we use 40-2 Wide Residual Network (Zagoruyko & Komodakis, 2016) models with normalization based on the CIFAR datasets and a dropout rate of 0.3. They are trained for 100 epochs with an initial learning rate of 0.1 that decreases following a cosine annealing schedule. Unless mentioned otherwise, each training step uses a batch of size 128 for the in-distribution and a batch of size 256 for the training out-distribution. The optimizer uses stochastic gradient descent with a Nesterov momentum of 0.9. Weight decay is set to $5 \cdot 10^{-4}$. The deep learning framework we use is PyTorch (Paszke et al., 2019), and for evaluating we use the scikit-learn (Pedregosa et al., 2011) implementation of the AUC. Our code is available at `https://anonymous.4open.science/r/OOD_BGC_BinDisc-D7FB`.

For evaluating the Mahalanobis detector, we use the code by the authors of Lee et al. (2018b) provided at `https://github.com/pokaxpoka/deep_Mahalanobis_detector`. The input noise levels and regression parameters are chosen on the available out-distribution OpenImages and are 0.0014 for CIFAR-10 and 0.002 for CIFAR-100.

All experiments were run on Nvidia V100 GPUs of an internal cluster of our institution, using up to 4 GB GPU memory (batch sizes in:128/out:256), with no noticeable difference between ours and the compared OE Hendrycks et al. (2019a) runs.

### C.2  DATASETS

We train our models with the train splits of CIFAR-10 and CIFAR-100 Krizhevsky & Hinton (2009) (MIT license) which each consist of 50,000 labeled images, and evaluate on their test splits of 10,000 samples. As training out-distribution we use OpenImages v4 (Krasin et al., 2017) (images have a CC BY 2.0 license); the training split that we employ here consists of 8,945,291 images of different sizes, which get resized to $32 \times 32$ pixels, and we test on 10,000 from the official validation split. For training with 80 Million Tiny Images (80M) (Torralba et al., 2008) in Appendix H (no license, see links in Appendix H), we use data from the beginning of the sequentialized dataset, and evaluate on a test set of 30,080 images starting at index 50,000,000. A subset of CIFAR images contained in 80M is excluded for training and evaluation. Further image datasets used for evaluation are SVHN (Netzer et al., 2011) (free for non-commercial use) with 26,032 samples, LSUN (Yu et al., 2015) Classroom (no license) with 300 samples, and CelebA (Liu et al., 2015) (available for non-commercial research purposes only) with 19,962 test samples. Uniform and Smooth Noise (Hein et al., 2019) are sampled, the latter by generating uniform noise and smoothing it using a Gaussian filter with a width that is drawn uniformly at random in $[1, 2.5]$. Each datapoint is then shifted and scaled linearly such that the minimal and maximal pixel values are 0 and 1, respectively. For both noises, we evaluate 30,080 inputs.

## D  RELATED WORK ON OOD DETECTION

Out-of-distribution detection has been an important research area in recent years, and several approaches that are fitted towards different training and inference scenarios have been proposed.

One seemingly obvious line of thought is to use generative models for density estimation to differentiate between in- and out-distribution (Bishop, 1994; Nalisnick et al., 2019; Ren et al., 2019; Nalisnick et al., 2019; Xiao et al., 2020). Recent methods to a certain extent overcome the problem mentioned in Nalisnick et al. (2019) that generative models can assign higher likelihood to distributions on which they have not been trained. Another line of work are score-based methods using an underlying classifier or the internal features of such a classifier, potentially combined with a generative model (Hendrycks & Gimpel, 2017; Liang et al., 2018; Lee et al., 2018c; Hendrycks et al., 2019a; Hein et al., 2019). One of the most effective methods up to now is Outlier Exposure  (Hendrycks et al., 2019a) and work building upon it (Chen et al., 2021; Meinke & Hein, 2020; Mohseni et al., 2020; Augustin et al., 2020; Papadopoulos et al., 2021; Thulasidasan et al., 2021) where a classifier is trained on

the in-distribution task and one enforces low confidence as proposed by Lee et al. (2018a) during training on a large and diverse set of out-of-distribution images (Hendrycks et al., 2019a) which can be seen as a proxy of all natural images. This approach generalizes well to other out-distributions. Recently, NTOM (Chen et al., 2021) has achieved excellent results for detecting far out-of-distribution data by adding a background class to the classifier which is trained on samples from the surrogate out-distribution that are mined such that they show a desired hardness for the model. At test time, the output probability for that class is used to decide if an input is to be flagged as OOD. Their ATOM method does the same while also adding adversarial perturbations to the OOD inputs during training. Even though it has been claimed that new approaches outperform Hendrycks et al. (2019a), up to our knowledge this has not been shown consistently across different and challenging test out-of-distribution datasets (including close and far out-of-distribution datasets). Below, we discuss some other recently proposed approaches that build upon different premises on the data available during training.

Hendrycks et al. (2019b) do not use any OOD data during training and instead teach the model to predict whether an input has been rotated by 0°, 90°, 180° or 270°. For inference, they use the loss of this predictor as an OOD score, and add this score to classifier output entropy, which behaves very similar to classifier confidence. Similar to our methods, they also use shared representations and the combination of the in-distribution classifier with a dedicated OOD detection score. If one interprets their rotation predictor loss as being an estimator of $\log p(o|x)$ for some implicit out-distribution, their scoring function coincides with our $s_2$ scores.

Golan & El-Yaniv (2018) learn a similar transformation detector (with Dirichlet statistics collected on the in-distribution replacing ground truth labels) and use it directly to detect OOD samples without using in-distribution class information.

Winkens et al. (2020) fit for each class a normalized Mahalanobis detector on the activations of a model trained with SimCLR ands a classification head on only the in-distribution with smoothed labels. They describe their method as applying class-wise density estimation in the feature space, where the normalized Mahalanobis distance is equivalent to a Gaussian density for each class.

Roy et al. (2021) treat an interesting application of flagging unseen skin diseases, making use of class labels that are also available for their training OOD data, which contains diseases that are different from both the in-distribution diseases and the unseen diseases. This allows them to do fine-grained OOD detection by regarding the sum over all OOD classes which for their dataset shows large improvement over methods that treat the training out-distributuion as one class. They gain additional slight improvements by combining this with a coarse grained binary loss that treats the sum over all in-distribution class probabilities as $p(i|x)$ and the sum over all OOD classes as $p(o|x)$. They show that this method can be combined with various representation learning approaches in order to improve their detection of unknown diseases.

Tack et al. (2020) introduce distribution shifting transformations into SimCLR training. Those are transformations that are so strong that the resulting samples can be considered as OOD and as negatives w.r.t. the original in the SimCLR loss. Similarly to Hendrycks et al. (2019b) and Golan & El-Yaniv (2018), they also train a head that classifies the applied transformation. In a version extended to using in-distribution labels, they consider samples from the same class and with the same transformation as positives, and samples where either is different as negatives. With this method, they obtain OOD detection results that significantly improve over standard classifier confidence, without using any training OOD dataset.

Liu & Abbeel (2020) derive a contrastive loss from Joint Energy Based Model training (Grathwohl et al., 2020) and train it together with cross-entropy on the in-distribution in order to obtain classifiers whose logit values can be transformed into energies that are equivalent as OOD scoring functions to in-distribution density. They show that using these energies yields some improvements over previous density estimation approaches, and that also the classifier confidences show moderately improved OOD detection when compared to standard training.

Liu et al. (2020) propose another energy based method which incorporates surrogate OOD data during training. We analyse this method in detail in Appendix F, where we show that their Bayes optimal OOD detector is equivalent to the binary discriminator between in- and out-distribution.

Li & Vasconcelos (2020) use the same training method as OE and show that careful resampling of the training out-distribution resembling hard negative mining can reduce its size and therefore lead to a more resource effective training OOD dataset, while the resulting models reach similarly good OOD detection performance.

# E    EXPERIMENTS COMPARING DENSITY ESTIMATION AND BINARY DISCRIMINATORS

## E.1    UNLABELED IN-DISTRIBUTION DATA AVAILABLE FOR TRAINING

We want to answer the following questions: How does estimating the in-distribution density compare to simply employing a binary discriminator between the in-distribution and uniform noise with respect to the task of out-of-distribution detection? Can other density based models be substituted for potentially easier to handle binary discriminators against a suitable (semi-)synthetic out-distribution? As generative models, we use a standard likelihood VAE, a likelihood PixelCNN++ and additionally compare with a Likelihood Regret VAE (Xiao et al., 2020). The binary classifier is trained to separate real data from uniform noise, thus none of the methods presented in this section make use of 80M or any other surrogate distribution. The results for OOD detection in terms of AUC for all methods are presented in Table 3.

Comparing the OOD detection performance of the binary discriminator trained against uniform noise with both VAE models, we asses that neither model is suitable for reliably detecting inputs from unknown out-distributions.

Following the theoretical analysis from the previous sections, the likelihood models and our binary classifier are able to perfectly separate the in-distribution data from uniform noise. This is expected as those methods are trained on that particular task of separating CIFAR-10 from uniform noise, whereas the LH Regret VAE with modified train objective has worse performance on uniform noise. It appears as if the training objective of the binary classifier seems to be too easy as the training and validation loss converge to almost zero in the first few epochs of training. However, the ability to separate uniform noise from real images does not generalize to other test distributions as both methods fail to achieve good out-of-distribution detection performance on the other test distributions. We note that while the score features from the likelihood models and the binary classifier are in expectation equivalent, both methods behave quite different on the test datasets (except for uniform noise). This is not surprising, as the probability of drawing real images from the uniform distribution is so small that neither training method properly regularizes the model's behaviour on those particular image manifolds. Thus the results are artifacts of random fluctuation and no method clearly outperforms the other one, for example the binary classifier is better at separating CIFAR-10 from SVHN whereas the likelihood VAE significantly outperforms the binary classifier on LSUN. Similar fluctuations also exist between the two variational auto encoders and PixelCNN++, but in conclusion none of those methods is able to generalize to more specialized unseen distributions.

Table 3: AUC for CIFAR-10 vs. various out-distributions of different methods that have access to only unlabelled CIFAR-10 data during training. Shown are the scores obtained from the likelihoods of the PixelCNN++ from Ren et al. (2019),

| MODEL | MEAN AUC | SVHN AUC | LSUN AUC | CELEBA AUC | SMOOTH AUC | C-100 AUC | OPEN AUC | 80M AUC | UNI AUC |
|---|---|---|---|---|---|---|---|---|---|
| LH PIXELCNN++ | 57.05 | 07.14 | 89.02 | 57.77 | 77.93 | 52.96 | 70.52 | 43.99 | 100.00 |
| LH VAE | **57.97** | 20.98 | 83.03 | 48.05 | 91.67 | 51.32 | 55.62 | 55.09 | 100.00 |
| LH REGRET VAE | 52.24 | 87.36 | 35.73 | 70.69 | 14.84 | 53.03 | 50.81 | 53.21 | 94.11 |
| BINDISC UNIFORM | 45.90 | 71.22 | 34.67 | 35.07 | 47.13 | 47.32 | 44.70 | 41.17 | 100.00 |

## E.2    LIKELIHOOD RATIOS AS A BINARY DISCRIMINATOR

In Section 3.1, we discussed that for the Bayes optimal solutions of their training objectives, the ratio of the likelihoods of two density estimators for different distributions is as an OOD detection scoring function equivalent to the prediction of a binary discriminator between the two distributions. In order to find out which role this equivalence plays in practise, we train a binary discriminator between CIFAR-10 as in-distribution and the background distribution obtained by mutating 10% of the pixles of in-distribution images as described in Ren et al. (2019). In Table 4, we compare the OOD detection performance of this discriminator with likelihood ratios estimated with PixelCNN++ (Salimans et al.,

2017) as trained with the code of Ren et al. (2019) setting $L_2$ regularization as 10, and with the numbers taken from Xiao et al. (2020) given for their VAE models. Even though we use the code and hyperparameter settings of Ren et al. (2019), the AUC we obtain for SVHN as out-distribution differs significantly from their reported 93.1%. We observe that all three methods struggle with detecting inputs from several out-distributions and thus we do not consider them as reliable out-of-distribution detection methods.

Table 4: AUROC for CIFAR-10 vs. various out-distributions. The Likelihood Ratio Ren et al. (2019) PixelCNN++ models were trained with their code. For the VAE, we cite the numbers of Xiao et al. (2020), as indicated by *. We also show a binary discriminator between CIFAR-10 and the background distribution of Ren et al. (2019).

| MODEL | MEAN | SVHN | LSUN | UNI | SMOOTH | C-100 | OPEN | CELEBA | 80M |
|---|---|---|---|---|---|---|---|---|---|
| LHRATIO PIXELCNN++ | **64.77** | 12.57 | 88.85 | 100.00 | 89.55 | 53.60 | 74.50 | 58.26 | 40.87 |
| LHRATIO VAE* | | 26.5* | 63.2* | 100.0* | | | | 44.7* | |
| BINDISC | 58.81 | 55.07 | 66.10 | 100.00 | 44.43 | 47.05 | 57.20 | 48.48 | 52.16 |

# F    ENERGY BASED OOD DETECTION IN TERMS OF BINARY DISCRIMINATORS

With Energy Based OOD Detection (Liu et al., 2020), we exhibit another surprising case of the equivalence to Binary Discriminators to an OOD detection method which is based on ideas that are quite different from those of the other extensively discussed methods. This method is based on the premise that the logits $f_\theta(x)$ of a classifier can be used to define an energy (here we ignore a potential temperature factor which as Liu et al. (2020) find can in good conscience be set to one)

$$E_\theta(x) = -\log \sum_{l=1}^{K} e^{f_\theta(l|x)} \tag{19}$$

that the model assigns to the input, which has been proposed by LeCun et al. (2006); Grathwohl et al. (2020). Ideally, this energy would be equivalent to the probability density of the in-distribution via

$$\log p(x|in) = -E_\theta(x) - Z \quad \text{where } Z = \int_{z \in [0,1]^D} e^{-E_\theta(z)} dz . \tag{20}$$

However, since the integral over the whole image domain is intractible, it is not possible to effectively decrease $-E_\theta$ on the in-distribution directly while also controlling $Z$. Naively training the classifier to have high energy on random inputs which would mean uniform noise is of course not a solution, since the model easily distinguishes the noise, and it is very unlikely to encounter any at least vaguely real images within finite time. Thus, Liu et al. (2020) rather use a surrogate training out-distribution of natural images which they increase the energy on during training; for their experiments, they take 80 Million Tiny Images, for which we compare their models with several other methods in Appendix H. Simultaneously, they minimize the standard classifier cross-entropy on the in-distribution. In order to avoid infinitely small potential losses, their training objective uses two margin hyper-parameters $m_{\text{in}} < m_{\text{out}}$ and reads

$$\mathbb{E}_{(x,y) \sim p(x,y|i)} -\log p_\theta(y|x)$$
$$+ \lambda \cdot \left( \mathbb{E}_{(x) \sim p(x|i)} \max\{0, E_\theta(x) - m_{\text{in}}\}^2 + \mathbb{E}_{w \sim p(w|o)} \max\{0, m_{\text{out}} - E_\theta(w)\}^2 \right) \tag{21}$$

with

$$\log p_\theta(y|x) = \log \frac{e^{f_\theta(y|x)}}{\sum_{l=1}^{K} e^{f_\theta(l|x)}} = f_\theta(y|x) + E_\theta(x) . \tag{22}$$

Note that their $\lambda$ does not balance between in and out by depending on a prior on $p(i)$, but it balances the energy regularization compared to the classifier cross-entropy loss.

**Theorem 3.** *The Bayes optimal logit output $f_\theta^*(x)$ of the energy based OOD detection model minimizing the expected loss on an input $x$ yields class probabilities $p_\theta^*(k|x) = p(k|x, i)$ that are optimal for a standard classifier with cross-entropy loss and simultaneously fulfills*

$$-E_\theta^*(x) = p(i|x) \cdot (m_{out} - m_{in}) - m_{out} . \tag{23}$$

*Proof.* Our goal is to find for a given input $x$ the model output $f_\theta(x)$ that minimizes the expected loss, assuming that we know the probabilities $p(y|x, i)$ and $p(i|x)$. This expected loss is

$$-p(i|x) \cdot \sum_{k=1}^{K} p(k|x, i) \log p_\theta(k|x) \tag{24}$$

$$+ \lambda \cdot \left( p(i|x) \max\{0, E_\theta(x) - m_{\text{in}}\}^2 + (1 - p(i|x)) \max\{0, m_{\text{out}} - E_\theta(x)\}^2 \right) . \tag{25}$$

First, we note that if a certain $E_\theta^*(x)$ minimizes the expected energy loss to

$$\lambda \cdot \left( p(i|x) \max\{0, E_\theta^*(x) - m_{\text{in}}\}^2 + (1 - p(i|x)) \max\{0, m_{\text{out}} - E_\theta^*(x)\}^2 \right) , \tag{26}$$

and if some $f_\theta^\sharp(x)$ minimizes the expected CE loss (note that its minimization is independent of the positive factor $p(i|x)$) to

$$-\sum_{k=1}^{K} p(k|x,i) \log \frac{e^{f_\theta^\sharp(k|x)}}{\sum_{l=1}^{K} e^{f_\theta^\sharp(l|x)}} \ , \tag{27}$$

with corresponding $E_\theta^\sharp(x) = -\log \sum_{i=1}^{K} e^{f_\theta^\sharp(x)[i]}$, then the logit output with $-E_\theta^*(x) + E_\theta^\sharp(x)$ added to each component has the energy

$$E_\theta(x) = -\log \sum_{i=1}^{K} e^{f_\theta^\sharp(x)[i] - E_\theta^*(x) + E_\theta^\sharp(x)} = -\log \left( e^{-E_\theta^*(x) + E_\theta^\sharp(x)} \cdot \sum_{i=1}^{K} e^{f_\theta^\sharp(x)[i]} \right) \tag{28}$$

$$= E_\theta^*(x) - E_\theta^\sharp(x) - \log \sum_{i=1}^{K} e^{f_\theta^\sharp(x)[i]} = E_\theta^*(x) \ , \tag{29}$$

which means that this $f^*(x)$ minimizes the expected energy loss, and also fulfills

$$p_\theta(x)[y] = \frac{e^{f_\theta^\sharp(x)[y] - E_\theta^*(x) + E_\theta^\sharp(x)}}{\sum_{l=1}^{K} e^{f_\theta^\sharp(x)[l] - E_\theta^*(x) + E_\theta^\sharp(x)}} = \frac{e^{f_\theta^\sharp(x)[y]}}{\sum_{l=1}^{K} e^{f_\theta^\sharp(x)[l]}} \ , \tag{30}$$

i.e. has the same probability predictions as $f^\sharp$ and thus also minimizes the expected CE loss. This means that both can be optimized independently.

As we've seen before, the optimal $p_\theta(k|x)$ for the expected CE loss is $p_\theta(k|x) = p(k|x,i)$.

Thus we independently minimize the expected energy loss

$$p(i|x) \max\{0, E_\theta(x) - m_{\text{in}}\}^2 + (1 - p(i|x)) \max\{0, m_{\text{out}} - E_\theta(x)\}^2 \ . \tag{31}$$

The first derivative with respect to $E_\theta(x)$ is

$$\begin{cases} 2 \cdot (1 - p(i|x)) \cdot (E_\theta(x) - m_{\text{out}}) & \text{for } E_\theta(x) < m_{\text{in}} \ , \\ 2 \cdot (p(i|x) \cdot (E_\theta(x) - m_{\text{in}}) + (1 - p(i|x)) \cdot (E_\theta(x) - m_{\text{out}})) & \text{for } m_{\text{in}} < E_\theta(x) < m_{\text{out}} \ , \\ 2 \cdot p(i|x) \cdot (E_\theta(x) - m_{\text{in}}) & \text{for } m_{\text{out}} < E_\theta(x) \ . \end{cases} \tag{32}$$

which simplified is

$$\begin{cases} < 0 & \text{for } E_\theta(x) < m_{\text{in}} \ , \\ 2 \cdot (E_\theta(x) + p(i|x) \cdot (m_{\text{out}} - m_{\text{in}}) - m_{\text{out}}) & \text{for } m_{\text{in}} < E_\theta(x) < m_{\text{out}} \ , \\ > 0 & \text{for } m_{\text{out}} < E_\theta(x) \ . \end{cases} \tag{33}$$

Here, for simplicity we again make the reasonable assumption that in- and out-distribution have full support, i.e. $0 < p(i|x) < 1$. (If we do not want to make the full support assumptions, for $p(i|x) = 0$, any energy $\geq m_{\text{out}}$ would be optimal, and for $p(i|x) = 1$, any energy $\leq m_{\text{in}}$ would be optimal.)

A the margin $E_\theta(x) = m_{\text{in}}$, the derivatives for $E_\theta(x) < m_{\text{in}}$ and $m_{\text{in}} < E_\theta(x) < m_{\text{out}}$ coincide as $2 \cdot (1 - p(i|x)) \cdot (m_{\text{in}} - m_{\text{out}}) < 0$, which means that the minimizer is $> m_{\text{in}}$. Similarly, we see that the minimizer is $< m_{\text{out}}$.

As the second derivative is positive, by solving where the derivative is zero, we find that the optimal negative energy (which is the score which they use for OOD detection) is

$$-E_\theta^*(x) = p(i|x) \cdot (m_{\text{out}} - m_{\text{in}}) - m_{\text{out}} \ . \tag{34}$$

$\square$

This is a strictly monotonously increasing function in $p(i|x)$. By our Theorem 1, we conclude that the negative energy score they obtain from their method is (for the Bayes optimal model that minimizes the loss on the training distributions) equivalent to $p(i|x)$.

**Corollary 2.** *At their Bayes optima, the Energy Based OOD Detection criterion is equivalent to that of a binary discriminator between the training in- and out-distributions.*

Note that while the classifier and energy loss terms *can* be optimized independently, the model is trained for both tasks simultaneously, which means that similar synergies to those we observe in 2 might contribute to the good performance of this method. Our empirical evaluation of energy based models in Appendix H confirms that their similar behaviour to the binary discriminator between in- and training-out-distribution indeed holds in practice.

# G  EVALUATIONS WITH THE AUC DETECTION METRIC

Complementing the evaluations of the FPR@95TPR metric presented in the main paper, Tables 5 and 6 show the AUC (AUROC) values of the same models as in Tables 1 and 2, respectively. The observations on the strngths of the different methods and scoring functions discussed in Section 4 also hold for the AUC evaluations, which is not surprising since the AUC and FPR@95TPR metrics are closely related.

Table 5: Accuracy on the in-distribution (CIFAR-10/CIFAR-100) and **AUC** for various test out-distributions of the different OOD methods with OpenImages as training out-distribution for which the FPRs are shown in Table 1.

| | | | | IN-DISTRIBUTION: CIFAR-10 | | | | | | |
|---|---|---|---|---|---|---|---|---|---|---|
| MODEL | ACC. | MEAN AUC | SVHN AUC | LSUN AUC | UNI AUC | SMOOTH AUC | C-100 AUC | 80M AUC | CELA AUC | OPENIM AUC |
| PLAIN CLASSI | 95.16 | 91.85 | 93.52 | 92.94 | 97.04 | 92.84 | 89.61 | 91.30 | 85.70 | 84.81 |
| MAHALANOBIS | | 91.63 | 96.34 | 92.39 | 100.00 | 99.82 | 86.78 | 89.41 | 76.67 | 84.81 |
| OE | 95.06 | **97.28** | 98.49 | 99.99 | 99.99 | 99.99 | 90.03 | 92.53 | 99.91 | 99.43 |
| BGC $s_1$ | | 95.02 | 99.48 | 100.00 | 99.99 | 99.95 | 79.64 | 86.37 | 99.74 | 99.97 |
| BGC $s_2$ | 95.21 | 97.22 | 98.90 | 100.00 | 99.99 | 99.67 | 90.47 | 92.41 | 99.11 | 99.73 |
| BGC $s_3$ | 95.21 | 97.21 | 98.87 | 100.00 | 99.98 | 99.62 | 90.47 | 92.41 | 99.08 | 99.71 |
| SHARED BINDISC | | 92.51 | 98.77 | 100.00 | 99.89 | 99.93 | 68.34 | 80.81 | 99.80 | 99.95 |
| SHARED CLASSI | 95.28 | 95.49 | 96.10 | 98.60 | 99.06 | 96.09 | 90.09 | 92.35 | 96.18 | 93.57 |
| SHARED COMBI $s_2$ | 95.28 | 97.26 | 98.66 | 100.00 | 99.93 | 99.94 | 89.71 | 92.84 | 99.72 | 99.88 |
| SHARED COMBI $s_3$ | 95.28 | 97.26 | 98.62 | 100.00 | 99.93 | 99.94 | 89.75 | 92.85 | 99.71 | 99.88 |
| | | | | IN-DISTRIBUTION: CIFAR-100 | | | | | | |
| MODEL | ACC. | MEAN AUC | SVHN AUC | LSUN AUC | UNI AUC | SMOOTH AUC | C-10 AUC | 80M AUC | | OPENIM AUC |
| PLAIN CLASSI | 77.16 | 82.13 | 82.33 | 79.13 | 96.03 | 81.36 | 76.14 | 77.80 | | 75.80 |
| MAHALANOBIS | | 85.88 | 88.69 | 87.56 | 90.08 | 99.91 | 70.78 | 78.26 | | 77.51 |
| OE | 77.19 | 90.37 | 89.54 | 99.98 | 99.03 | 99.68 | 75.95 | 78.03 | | 99.67 |
| BGC $s_1$ | | 88.41 | 97.38 | 99.99 | 99.70 | 99.79 | 60.51 | 73.11 | | 99.93 |
| BGC $s_2$ | **77.61** | 90.47 | 90.50 | 99.99 | 99.87 | 99.75 | 74.88 | 77.82 | | 99.64 |
| BGC $s_3$ | **77.61** | 90.46 | 90.46 | 99.99 | 99.88 | 99.74 | 74.88 | 77.82 | | 99.64 |
| SHARED BINDISC | | 84.62 | 97.44 | 99.99 | 99.70 | 99.68 | 47.82 | 63.13 | | 99.93 |
| SHARED CLASSI | 77.35 | 82.06 | 82.72 | 99.05 | 72.73 | 84.14 | 75.76 | 77.99 | | 93.54 |
| SHARED COMBI $s_2$ | 77.35 | **90.74** | 91.74 | 99.99 | 99.59 | 99.54 | 75.50 | 78.10 | | 99.57 |
| SHARED COMBI $s_3$ | 77.35 | 90.73 | 91.69 | 99.99 | 99.57 | 99.53 | 75.50 | 78.10 | | 99.57 |

Table 6: **AUC** evaluation of the models trained with shared and separate representations from Table 2.

IN-DISTRIBUTION: CIFAR-10

| MODEL | ACC. | MEAN AUC | SVHN AUC | LSUN AUC | UNI AUC | SMOOTH AUC | C-100 AUC | 80M AUC | CELA AUC | OPENIM AUC |
|---|---|---|---|---|---|---|---|---|---|---|
| PLAIN CLASSI | 95.16 | 91.85 | 93.52 | 92.94 | 97.04 | 92.84 | 89.61 | 91.30 | 85.70 | 84.81 |
| SEPARATE BINDISC | | 89.03 | 96.42 | 100.00 | 99.97 | 99.99 | 58.60 | 72.36 | 95.87 | 99.99 |
| SEPARATE COMBI $s_3$ | 95.16 | 96.40 | 98.16 | 100.00 | 99.98 | 99.99 | 89.64 | 91.35 | 95.70 | 99.94 |
| SHARED BINDISC | | 92.51 | 98.77 | 100.00 | 99.89 | 99.93 | 68.34 | 80.81 | 99.80 | 99.95 |
| SHARED CLASSI | **95.28** | 95.49 | 96.10 | 98.60 | 99.06 | 96.09 | 90.09 | 92.35 | 96.18 | 93.57 |
| SHARED COMBI $s_3$ | **95.28** | 97.26 | 98.62 | 100.00 | 99.93 | 99.94 | 89.75 | 92.85 | 99.71 | 99.88 |
| PLAIN $\otimes$ SHA DISC $s_3$ | 95.16 | **97.26** | 98.67 | 100.00 | 99.91 | 99.93 | 89.78 | 92.95 | 99.58 | 99.87 |

IN-DISTRIBUTION: CIFAR-100

| MODEL | ACC. | MEAN | SVHN | LSUN | UNI | SMOOTH | C-10 | OPENIM | 80M |
|---|---|---|---|---|---|---|---|---|---|
| PLAIN CLASSI | 77.16 | 82.13 | 82.33 | 79.13 | 96.03 | 81.36 | 76.14 | 77.80 | 75.80 |
| SEPARATE BINDISC | | 84.30 | 94.68 | 100.00 | 99.81 | 99.64 | 50.06 | 61.62 | 99.98 |
| SEPARATE COMBI $s_3$ | 77.16 | 88.95 | 84.49 | 99.99 | 99.88 | 96.06 | 75.83 | 77.45 | 99.82 |
| SHARED BINDISC | | 84.62 | 97.44 | 99.99 | 99.70 | 99.68 | 47.82 | 63.13 | 99.93 |
| SHARED CLASSI | **77.35** | 82.06 | 82.72 | 99.05 | 72.73 | 84.14 | 75.76 | 77.99 | 93.54 |
| SHARED COMBI $s_3$ | **77.35** | 90.73 | 91.69 | 99.99 | 99.57 | 99.53 | 75.50 | 78.10 | 99.57 |
| PLAIN $\otimes$ SHA DISC $s_3$ | 77.16 | **90.79** | 91.99 | 99.99 | 99.80 | 98.85 | 75.87 | 78.23 | 99.50 |

## H    DISCUSSION OF THE CHOICE OF OPENIMAGES AND EXPERIMENTAL RESULTS WITH 80 MILLION TINY IMAGES AS TRAINING OUT-DISTRIBUTION

Since the 80 Million Tiny Images (Torralba et al., 2008) dataset was retracted by the authors – their statement can be read at `http://groups.csail.mit.edu/vision/TinyImages/`[1] – as a reaction to Birhane & Prabhu (2021) exposing the presence of offensive and prejudicial images in the dataset, a good "surrogate surrogate" training out-distribution for the CIFAR-10/CIFAR-100 in-distribution has to our knowledge not yet been established.

Our experience confirms the assessment which the authors of Hendrycks et al. (2019a) make in their discussion section 5: the surrogate training out-distribution should consist mainly of natural images, should have a high semantic diversity, and the number of samples in the dataset should be large. We also observe that it is vital that there are no easy to detect details that separate the training in- and out-distributions from each other, as for example using a different resizing interpolation method would lead to 'overfitting' on such features with 100% train accuracy of the Binary Discriminator and near zero loss (apart from that of the in-distribution classifier) for OE and BGC, with no generalization to the test OOD datasets. OpenImages fulfills the mentioned criteria, and in our judgement does not contain ethically problematic images. It is, however, to be noted that CIFAR was sourced as a subset of 80M Tiny Images, see Krizhevsky & Hinton (2009) and `https://www.cs.toronto.edu/~kriz/cifar.html`, which explains the somewhat better results with 80M as training OOD dataset that we observe below in this section. Our results on the theoretical relations between the different methods and scoring functions do not depend on the choice of the training OOD dataset , and give reason to expect that our experimental confirmations of these relations will also hold for even better suited surrogate OOD datasets that we hope will be found in future works.

In the main paper, we employ OpenImages (Krasin et al., 2017) as a replacement, and for completeness and comparison to the originals of OE, Energy Based OOD detection and NTOM/ATOM, below we show and discuss the results obtained with the retracted 80M dataset which is commonly used in OOD detection literature.

The training procedure of our methods again follows that of OE, and for the 80M experiments we do not add AutoAugment, in order to stay as close as possible to the original. For the established methods we compare to, we use the original weights of their published models; the Plain and Outlier Exposure (OE) models were retrieved from the repository of the authors of OE (`https://github.com/hendrycks/outlier-exposure`), and for NTOM and ATOM (Chen et al., 2021) we use their code `https://github.com/jfc43/informative-outlier-mining` to evaluate their DenseNet models (their best models). We finetune the Mahalanobis detector on 80M with the same procedure as described above; the optimal input noise level for 80M is 0.0005 for both CIFAR in-distributions. We evaluate the Energy Based models fine-tuned on 80M which the authors of Liu et al. (2020) provide at `https://github.com/wetliu/energy_ood` with their evaluation code.

For CIFAR-10, as already seen with OpenImages as training out-distribution, the OOD detection performance of SHARED CLASSI is much better than that of the plain classifier. .In fact SHARED BINDISC has already very good OOD performance with a mean FPR of 7.56 and mean AUC of 97.90 which is only improved by considering scoring function $s_2/s_3$ in the combination of SHARED BINDISC and SHARED CLASSI which yields the best performance in classification accuracy and mean AUC.

The classifier with 80M as background class (BGC) works very well for all scoring functions and reaches SOTA performance similar to/better than OE. Both BGC/SHARED COMBI with $s_2/s_3$ perform particularly well on the challenging close out-distributions CIFAR-100 and OpenImages (which are the data sets where NTOM/ATOM perform significantly worse). However, as already observed with OpenImages as training out-distribution the differences of the methods are minor both in terms of classification accuracy.

---

[1]Archived statement: `https://web.archive.org/web/20210415160225/http://groups.csail.mit.edu/vision/TinyImages/`

The results for CIFAR-100 are again qualitatively similar to those for CIFAR-10. NTOM/ATOM now show worse mean AUC results which are mainly due to worse results for the close out-distributions CIFAR-10 and OpenImages, but better mean FPR@95%TPR, which is explained by their excellent detection of LSUN Classroom, where the other methods work quite well in terms of AUC but still make quite many errors at the 95%TPR threshold. OE again achieves comparable OOD results to the other evaluated methods. As the theoretical considerations we presented in Appendix F suggest, the energy based OOD detector achieves good performance that is comparable that to the other methods. Our SHARED COMBI $s_2/s_3$ performs best in terms of OOD performance and test accuracy but again differences are minor.

The conclusions are similar to those drawn for OpenImages in the main paper. Comparing the results, it is clear that 80M still works somewhat better than OpenImages, so for the CIFAR in-distributions, the search for an ethically acceptable replacement of 80M that allows for equal or improved results continues. The consistent similarities between the examined methods over the different datasets suggest that the methods and scoring functions would be similarly viable with such an alternative training OOD dataset.

Table 7: Accuracy on the in-distribution (CIFAR-10/CIFAR-100) and **FPR@95%TPR** for various test out-distributions of different OOD methods with **80 Million Tiny Images** as training out-distribution (shown results for test set of 80M are not used for computing the mean FPR). Lower false positive rate is better. CelebA makes no sense as test out-distribution for CIFAR-100 as it contains man/woman as classes. PLAIN, OE, BGC and SHARED have been trained using the same architecture and training parameters/schedule. $s_1, s_2, s_3$ are the scoring functions introduced in Section 3.2. Our binary discriminator (BINDISC) resp. the combination with the shared classifier (SHARED COMBI) performs similar/better than Outlier Exposure (Hendrycks et al., 2019a).

IN-DISTRIBUTION: CIFAR-10

| MODEL | ACC. | MEAN FPR | SVHN FPR | LSUN FPR | UNI FPR | SMOOTH FPR | C-100 FPR | OPENIM FPR | CELA FPR | 80M FPR |
|---|---|---|---|---|---|---|---|---|---|---|
| PLAIN CLASSI | 94.84 | 64.62 | 48.33 | 52.67 | 75.69 | 62.58 | 62.91 | 66.38 | 83.79 | 60.53 |
| MAHALANOBIS | | 41.15 | 40.19 | 50.00 | 0.00 | 0.17 | 58.66 | 58.36 | 80.69 | 53.42 |
| ENERGY | 95.22 | 9.01 | 1.58 | 2.00 | 0.00 | 0.00 | 30.03 | 28.26 | 1.19 | 8.21 |
| NTOM | 95.42 | 8.21 | 1.06 | 0.33 | 0.00 | 0.00 | 29.61 | 26.15 | 0.30 | 4.90 |
| ATOM | 95.20 | 7.76 | 0.69 | 0.33 | 0.00 | 0.00 | 27.80 | 25.26 | 0.25 | 4.44 |
| OE | 95.74 | 8.27 | 1.96 | 2.00 | 0.00 | 0.06 | 26.12 | 27.07 | 0.71 | 5.96 |
| BGC $s_1$ | | 7.47 | 0.83 | 1.33 | 0.00 | 0.00 | 24.75 | 25.19 | 0.19 | 4.43 |
| BGC $s_2$ | 95.63 | **7.42** | 0.98 | 1.33 | 0.00 | 0.00 | 24.13 | 25.33 | 0.20 | 4.95 |
| BGC $s_3$ | 95.63 | 7.49 | 1.05 | 1.33 | 0.00 | 0.00 | 24.26 | 25.57 | 0.21 | 4.82 |
| SHARED BINDISC | | 7.56 | 0.67 | 1.67 | 0.00 | 0.00 | 24.70 | 25.58 | 0.31 | 4.57 |
| SHARED CLASSI | **96.08** | 15.71 | 6.29 | 13.00 | 0.07 | 0.13 | 37.07 | 40.48 | 12.95 | 19.47 |
| SHARED COMBI $s_2$ | **96.08** | 7.47 | 0.71 | 1.33 | 0.00 | 0.00 | 24.15 | 25.72 | 0.35 | 4.79 |
| SHARED COMBI $s_3$ | **96.08** | 7.44 | 0.73 | 1.33 | 0.00 | 0.00 | 23.95 | 25.70 | 0.35 | 4.84 |

IN-DISTRIBUTION: CIFAR-100

| MODEL | ACC. | MEAN FPR | SVHN FPR | LSUN FPR | UNI FPR | SMOOTH FPR | C-10 FPR | OPENIM FPR | 80M FPR |
|---|---|---|---|---|---|---|---|---|---|
| PLAIN CLASSI | 75.96 | 82.26 | 84.33 | 80.00 | 98.99 | 65.81 | 81.97 | 82.47 | 80.17 |
| MAHALANOBIS | | 47.89 | 64.58 | 63.67 | 0.00 | 2.77 | 81.39 | 74.93 | 69.79 |
| ENERGY | 75.70 | 32.95 | 20.61 | 16.67 | 4.23 | 2.90 | 84.27 | 69.00 | 42.18 |
| NTOM | 74.88 | **32.63** | 24.67 | 10.00 | 0.00 | 0.00 | 90.58 | 70.52 | 40.78 |
| ATOM | 75.06 | 34.60 | 37.78 | 8.67 | 0.00 | 0.30 | 89.80 | 71.02 | 40.29 |
| OE | **76.73** | 34.89 | 34.41 | 24.00 | 1.10 | 4.96 | 79.77 | 65.09 | 45.59 |
| BGC $s_1$ | | 34.79 | 35.73 | 23.00 | 0.00 | 0.07 | 81.61 | 68.31 | 45.76 |
| BGC $s_2$ | 75.82 | 35.86 | 40.36 | 26.67 | 0.00 | 0.45 | 79.50 | 68.21 | 47.72 |
| BGC $s_3$ | 75.82 | 35.91 | 40.53 | 26.67 | 0.00 | 0.42 | 79.54 | 68.28 | 47.46 |
| SHARED BINDISC | | 32.74 | 25.16 | 22.00 | 0.00 | 0.00 | 82.47 | 66.83 | 44.50 |
| SHARED CLASSI | 76.52 | 44.86 | 56.96 | 57.00 | 0.06 | 0.22 | 79.55 | 75.37 | 65.50 |
| SHARED COMBI $s_2$ | 76.52 | 32.71 | 25.70 | 23.67 | 0.00 | 0.00 | 80.95 | 65.93 | 44.87 |
| SHARED COMBI $s_3$ | 76.52 | 32.72 | 25.79 | 23.67 | 0.00 | 0.00 | 80.90 | 65.98 | 44.96 |

Table 8: Accuracy on the in-distribution (CIFAR-10/CIFAR-100) and **AUC** (AUROC) for various test out-distributions of different OOD methods with **80 Million Tiny Images** as training out-distribution (shown results for test set of 80M are not used for computing the mean AUC). The relative performance of the different methods measured in AUC is similar to what we observed with the FPR@95%TPR measure in Table 7.

IN-DISTRIBUTION: CIFAR-10

| MODEL | ACC. | MEAN AUC | SVHN AUC | LSUN AUC | UNI AUC | SMOOTH AUC | C-100 AUC | OPENIM AUC | CELA AUC | 80M AUC |
|---|---|---|---|---|---|---|---|---|---|---|
| PLAIN CLASSI | 94.84 | 85.75 | 91.91 | 91.63 | 87.69 | 78.27 | 87.83 | 83.23 | 79.43 | 88.01 |
| MAHALANOBIS | | 91.12 | 94.34 | 91.98 | 100.00 | 99.51 | 88.08 | 84.92 | 79.17 | 89.65 |
| ENERGY | 95.22 | 97.32 | 99.26 | 99.49 | 99.00 | 99.40 | 93.81 | 90.73 | 99.57 | 97.71 |
| NTOM | 95.42 | 97.32 | 99.59 | 99.79 | 99.97 | 99.84 | 92.19 | 89.96 | 99.89 | 98.72 |
| ATOM | 95.20 | 97.42 | 99.63 | 99.76 | 99.93 | 99.60 | 92.89 | 90.30 | 99.85 | 98.55 |
| OE | 95.74 | 97.64 | 99.48 | 99.48 | 99.46 | 99.64 | 94.80 | 90.91 | 99.71 | 98.50 |
| BGC $s_1$ | | 97.94 | 99.64 | 99.58 | 99.96 | 99.98 | 94.84 | 91.65 | 99.92 | 98.78 |
| BGC $s_2$ | 95.63 | 97.95 | 99.60 | 99.52 | 99.97 | 99.98 | 95.03 | 91.65 | 99.92 | 98.65 |
| BGC $s_3$ | 95.63 | 97.95 | 99.58 | 99.52 | 99.97 | 99.98 | 95.04 | 91.64 | 99.92 | 98.70 |
| SHARED BINDISC | | 97.90 | 99.74 | 99.60 | 99.94 | 99.96 | 94.75 | 91.42 | 99.87 | 98.77 |
| SHARED CLASSI | **96.08** | 96.57 | 98.77 | 97.78 | 99.87 | 99.68 | 93.40 | 88.70 | 97.80 | 96.42 |
| SHARED COMBI $s_2$ | **96.08** | **97.96** | 99.73 | 99.58 | 99.95 | 99.96 | 95.13 | 91.48 | 99.85 | 98.73 |
| SHARED COMBI $s_3$ | **96.08** | **97.96** | 99.73 | 99.58 | 99.95 | 99.96 | 95.14 | 91.48 | 99.85 | 98.73 |

IN-DISTRIBUTION: CIFAR-100

| MODEL | ACC. | MEAN AUC | SVHN AUC | LSUN AUC | UNI AUC | SMOOTH AUC | C-10 AUC | OPENIM AUC | 80M AUC |
|---|---|---|---|---|---|---|---|---|---|
| PLAIN CLASSI | 75.96 | 77.48 | 71.38 | 76.89 | 78.14 | 88.36 | 75.33 | 74.60 | 75.92 |
| MAHALANOBIS | | 87.75 | 85.96 | 87.20 | 100.00 | 99.16 | 75.45 | 78.74 | 79.76 |
| ENERGY | 75.70 | 91.67 | 96.54 | 96.69 | 97.91 | 98.92 | 77.39 | 82.57 | 91.16 |
| NTOM | 74.88 | 88.49 | 96.20 | 97.31 | 99.79 | 99.94 | 62.44 | 75.24 | 88.41 |
| ATOM | 75.06 | 88.02 | 93.68 | 97.51 | 99.98 | 98.46 | 63.47 | 75.02 | 88.44 |
| OE | **76.73** | 91.72 | 94.06 | 95.58 | 99.06 | 98.84 | 79.53 | 83.31 | 88.43 |
| BGC $s_1$ | | **92.04** | 94.42 | 95.47 | 99.99 | 99.73 | 79.15 | 83.46 | 89.19 |
| BGC $s_2$ | 75.82 | 91.54 | 93.32 | 94.64 | 99.95 | 99.63 | 79.29 | 82.41 | 88.11 |
| BGC $s_3$ | 75.82 | 91.53 | 93.30 | 94.62 | 99.94 | 99.62 | 79.29 | 82.40 | 88.23 |
| SHARED BINDISC | | 91.84 | 95.90 | 95.69 | 99.79 | 99.94 | 76.56 | 83.19 | 89.25 |
| SHARED CLASSI | 76.52 | 88.16 | 86.28 | 87.61 | 99.97 | 99.90 | 77.00 | 78.23 | 81.72 |
| SHARED COMBI $s_2$ | 76.52 | 92.03 | 95.50 | 95.43 | 99.96 | 99.98 | 78.46 | 82.86 | 88.69 |
| SHARED COMBI $s_3$ | 76.52 | 92.03 | 95.49 | 95.42 | 99.97 | 99.98 | 78.46 | 82.85 | 88.67 |

## I  EXPERIMENTS WITH SVHN AS TRAINING OUT-DISTRIBUTION

In order to examine the effect of a training out-distribution that relatively far away from the in-distribution, we show experiments with SVHN as out-distribution in Tables 9 and 10. The OOD detection performance of these methods is much worse than with the closer OpenImages and 80M training out-distributions. In most cases, combinations $s_2$ and $s_3$/OE which implicitly or explicitly use the classifier confidence lead to better OOD detection than using the binary discriminator/$s_1$. The inconsistent behaviour over the different test out.distributions of the methods that use SVHN as training out-distribution can be explained by the easiness of the discrimination task, which manifests itself in the fact that $BGCs_1$ and SHARED BINDISC reach perfect FPR and AUC metrics. This indicates a form of overfitting to this specific out-distribution, without consistent generalization to unseen distributions which do not have characteristic features that are similar to those appearing in SVHN images.

We also do not observe a beneficial effect on test accuracy for the methods that use SVHN compared to plain, which is to be expected as the representations learned from SVHN are hardly useful for the in-distribution task.

Table 9: **FPR@95%TPR** for CIFAR-10/CIFAR-100 as in-distribution with **SVHN** as training out-distribution.

| | | | | | IN-DISTRIBUTION: CIFAR-10 | | | | | |
|---|---|---|---|---|---|---|---|---|---|---|
| MODEL | ACC. | MEAN FPR | LSUN FPR | UNI FPR | SMOOTH FPR | C-100 FPR | 80M FPR | OPENIM FPR | CELA FPR | SVHN FPR |
| PLAIN CLASSI | **94.84** | 64.62 | 48.33 | 52.67 | 75.69 | 62.58 | 62.91 | 66.38 | 83.79 | 60.53 |
| OE | 94.80 | 63.73 | 54.33 | 92.01 | 41.27 | 60.55 | 55.91 | 65.52 | 76.51 | 0.03 |
| BGC $s_1$ | | 54.31 | 53.00 | 100.00 | 0.51 | 55.16 | 39.28 | 64.36 | 67.87 | 0.00 |
| BGC $s_2$ | 94.51 | 62.98 | 58.33 | 98.53 | 18.73 | 62.27 | 57.47 | 67.35 | 78.16 | 0.02 |
| BGC $s_3$ | 94.51 | 63.08 | 58.33 | 98.52 | 19.18 | 62.34 | 57.63 | 67.39 | 78.16 | 0.02 |
| SHARED BINDISC | | 72.95 | 95.67 | 100.00 | 5.81 | 77.04 | 66.75 | 85.17 | 80.22 | 0.00 |
| SHARED CLASSI | 94.71 | 48.39 | 54.33 | 0.00 | 17.99 | 62.34 | 57.37 | 67.05 | 79.64 | 0.57 |
| SHARED COMBI $s_2$ | 94.71 | **46.27** | 55.00 | 0.00 | 6.74 | 60.84 | 55.15 | 66.50 | 79.65 | 0.02 |
| SHARED COMBI $s_3$ | 94.71 | 46.31 | 55.00 | 0.00 | 6.84 | 60.90 | 55.26 | 66.53 | 79.65 | 0.02 |

| | | | | | IN-DISTRIBUTION: CIFAR-100 | | | | |
|---|---|---|---|---|---|---|---|---|---|
| MODEL | ACC. | MEAN FPR | LSUN FPR | UNI FPR | SMOOTH FPR | C-10 FPR | 80M FPR | OPENIM FPR | SVHN FPR |
| PLAIN CLASSI | 75.96 | 82.26 | 84.33 | 80.00 | 98.99 | 65.81 | 81.97 | 82.47 | 80.17 |
| OE | 75.78 | **73.89** | 81.33 | 99.47 | 15.93 | 83.32 | 80.30 | 83.01 | 0.04 |
| BGC $s_1$ | | 77.16 | 96.33 | 100.00 | 3.35 | 91.05 | 80.55 | 91.68 | 0.00 |
| BGC $s_2$ | 75.21 | 74.53 | 80.00 | 98.49 | 21.49 | 83.14 | 80.52 | 83.54 | 0.07 |
| BGC $s_3$ | 75.21 | 74.53 | 80.00 | 98.49 | 21.50 | 83.14 | 80.53 | 83.54 | 0.07 |
| SHARED BINDISC | | 82.08 | 99.33 | 100.00 | 10.04 | 95.00 | 91.21 | 96.89 | 0.00 |
| SHARED CLASSI | **75.97** | 83.46 | **78.67** | 100.00 | 76.67 | 82.36 | 79.99 | 83.08 | 0.44 |
| SHARED COMBI $s_2$ | **75.97** | 78.66 | **78.67** | 100.00 | 47.73 | 82.44 | 79.93 | 83.20 | 0.07 |
| SHARED COMBI $s_3$ | **75.97** | 78.66 | 78.67 | 100.00 | 47.78 | 82.43 | 79.92 | 83.17 | 0.07 |

Table 10: **AUC** for CIFAR-10/CIFAR-100 as in-distribution with **SVHN** as training out-distribution.

IN-DISTRIBUTION: CIFAR-10

| MODEL | ACC. | MEAN AUC | LSUN AUC | UNI AUC | SMOOTH AUC | C-100 AUC | 80M AUC | OPENIM AUC | CELA AUC | SVHN AUC |
|---|---|---|---|---|---|---|---|---|---|---|
| PLAIN CLASSI | **94.84** | 85.75 | 91.91 | 91.63 | 87.69 | 78.27 | 87.83 | 83.23 | 79.43 | 88.01 |
| OE | 94.80 | 88.37 | 91.72 | 91.33 | 91.20 | 88.26 | 89.55 | 82.97 | 83.59 | 100.00 |
| BGC $s_1$ | | 83.58 | 89.43 | 58.34 | 99.80 | 85.44 | 90.28 | 78.01 | 83.76 | 100.00 |
| BGC $s_2$ | 94.51 | 89.62 | 91.50 | 91.14 | 97.60 | 88.72 | 89.98 | 82.96 | 85.40 | 100.00 |
| BGC $s_3$ | 94.51 | 89.60 | 91.50 | 91.14 | 97.54 | 88.71 | 89.96 | 82.96 | 85.40 | 100.00 |
| SHARED BINDISC | | 58.47 | 37.27 | 38.61 | 98.85 | 55.19 | 65.66 | 48.61 | 65.12 | 100.00 |
| SHARED CLASSI | 94.71 | 90.14 | 90.20 | 99.70 | 97.44 | 88.50 | 89.68 | 82.97 | 82.51 | 99.80 |
| SHARED COMBI $s_2$ | 94.71 | 90.18 | 89.74 | 99.67 | 99.00 | 88.49 | 89.87 | 82.43 | 82.09 | 100.00 |
| SHARED COMBI $s_3$ | 94.71 | **90.19** | 89.76 | 99.68 | 98.97 | 88.50 | 89.87 | 82.46 | 82.12 | 100.00 |

IN-DISTRIBUTION: CIFAR-100

| MODEL | ACC. | MEAN AUC | LSUN AUC | UNI AUC | SMOOTH AUC | C-10 AUC | 80M AUC | OPENIM AUC | SVHN AUC |
|---|---|---|---|---|---|---|---|---|---|
| PLAIN CLASSI | 75.96 | 77.48 | 71.38 | 76.89 | 78.14 | 88.36 | 75.33 | 74.60 | 75.92 |
| OE | 75.78 | 78.86 | 75.93 | 73.53 | 97.02 | 75.37 | 76.57 | 74.74 | 99.99 |
| BGC $s_1$ | | 66.60 | 67.45 | 24.51 | 99.37 | 68.50 | 73.82 | 65.97 | 100.00 |
| BGC $s_2$ | 75.21 | 77.38 | 76.04 | 67.25 | 96.28 | 74.83 | 75.86 | 74.01 | 99.98 |
| BGC $s_3$ | 75.21 | 77.38 | 76.04 | 67.25 | 96.27 | 74.83 | 75.86 | 74.01 | 99.98 |
| SHARED BINDISC | | 55.60 | 34.81 | 53.23 | 98.03 | 49.00 | 54.57 | 43.93 | 100.00 |
| SHARED CLASSI | **75.97** | 72.04 | 80.31 | 41.57 | 83.32 | 75.64 | 76.58 | 74.81 | 99.90 |
| SHARED COMBI $s_2$ | **75.97** | 72.95 | 80.23 | 41.34 | 89.21 | 75.59 | 76.57 | 74.75 | 99.99 |
| SHARED COMBI $s_3$ | **75.97** | 72.94 | 80.23 | 41.34 | 89.18 | 75.59 | 76.57 | 74.75 | 99.99 |

## J    EXPERIMENTS WITH RESTRICTED IMAGENET AS IN-DISTRIBUTION

In addition to the results for CIFAR-10 and CIFAR-100 shown in the main paper, here we provide results for Restricted ImageNet. Restricted ImageNet, introduced by Tsipras et al. (2019), consists of 9 classes, where each individual class is a union of multiple ImageNet (Deng et al., 2009) classes, for example the Restricted ImageNet class 'dog' contains all dog breeds from ImageNet. As Restricted ImageNet only contains animal classes, the union over all its classes does not cover the entire ILSVRC2012 dataset (Russakovsky et al., 2015), which allows us to use the remaining ILSVRC2012 classes as training out-distribution. Like we did for the CIFAR experiments, we train a plain classifier, an Outlier Exposure model, a background class model and a shared discriminator/classifier and evaluate them with the different scoring functions. The model is a ResNet50 and we use random cropping and flipping as data augmentation during training. The results in terms of FPR@95%TPR and AUC can be found in Table 11.

Once again, we see that SHARED CLASSI has relatively good OOD detection performance and clearly beats the plain classifier from standard training. Again, we see that Outlier Exposure Hendrycks et al. (2019a), training with background class and shared training of classifier and binary discriminator perform similarly. In terms of accuracy, the shared model benefits most from the added unlabelled data compared to plain training. At 95% TPR, the sharedly trained binary discriminator and its combinations SHARED COMBI $s_2$ and SHARED COMBI $s_2$ detect flower images significantly better than the other approaches which results in the best mean FPR@95%TPR compared to the other methods, while in terms of AUC, OE has a slight advantage.

Table 11: Out-of-distribution detection evaluation for various ResNet50 models trained on **Restricted ImageNet** in terms of AUC and FPR@95%TPR. The last column (NOTRIN) refers to the remaining classes from the ILSVRC2012 validation split that are not part of Restricted ImageNet and that were used as the training out-distribution;it does not contribute to the mean test FPR/AUC. As all models use the train split of NotRIN as training-out distribution.

IN-DISTRIBUTION: RESTRICTED IMAGENET

FPR@95%TPR

| MODEL | ACC. | MEAN FPR | FLOWERS FPR | FGVC FPR | CARS FPR | SMOOTH FPR | UNIFORM FPR | NOTRIN FPR |
|---|---|---|---|---|---|---|---|---|
| PLAIN CLASSI | 96.34 | 36.71 | 60.11 | 50.23 | 73.20 | 0.00 | 0.00 | 50.37 |
| OE | 97.10 | 4.26 | 21.06 | 0.18 | 0.04 | 0.00 | 0.00 | 6.91 |
| BGC $s_1$ | | 4.22 | 20.74 | 0.33 | 0.01 | 0.00 | 0.00 | 6.46 |
| BGC $s_2$ | 97.50 | 4.77 | 23.43 | 0.39 | 0.02 | 0.00 | 0.00 | 6.02 |
| BGC $s_3$ | 97.50 | 4.73 | 23.26 | 0.36 | 0.01 | 0.00 | 0.00 | 6.13 |
| SHARED BINDISC | | 2.73 | 10.10 | 0.24 | 0.04 | 3.26 | 0.00 | 5.93 |
| SHARED CLASSI | **97.59** | 17.51 | 50.54 | 17.79 | 19.21 | 0.00 | 0.00 | 21.99 |
| SHARED COMBI $s_2$ | **97.59** | **2.55** | 12.28 | 0.45 | 0.04 | 0.00 | 0.00 | 5.58 |
| SHARED COMBI $s_3$ | **97.59** | 2.62 | 12.60 | 0.45 | 0.04 | 0.00 | 0.00 | 5.63 |

AUC

| MODEL | ACC. | MEAN AUC | FLOWERS AUC | FGVC AUC | CARS AUC | SMOOTH AUC | UNIFORM AUC | NOTRIN AUC |
|---|---|---|---|---|---|---|---|---|
| PLAIN CLASSI | 96.34 | 94.96 | 91.65 | 92.67 | 92.46 | 98.74 | 99.26 | 92.38 |
| OE | 97.10 | **98.76** | 96.65 | 99.75 | 99.85 | 97.95 | 99.58 | 98.46 |
| BGC $s_1$ | | 98.61 | 96.64 | 99.86 | 99.97 | 97.77 | 98.80 | 98.67 |
| BGC $s_2$ | 97.50 | 98.66 | 96.39 | 99.83 | 99.96 | 98.18 | 98.94 | 98.69 |
| BGC $s_3$ | 97.50 | 98.66 | 96.43 | 99.83 | 99.96 | 98.14 | 98.93 | 98.68 |
| SHARED BINDISC | | 98.26 | 97.62 | 99.83 | 99.94 | 96.13 | 97.78 | 98.71 |
| SHARED CLASSI | **97.59** | 96.93 | 93.40 | 96.58 | 96.53 | 99.48 | 98.66 | 96.10 |
| SHARED COMBI $s_2$ | **97.59** | 98.54 | 97.41 | 99.80 | 99.93 | 97.37 | 98.18 | 98.72 |
| SHARED COMBI $s_3$ | **97.59** | 98.58 | 97.36 | 99.79 | 99.92 | 97.61 | 98.22 | 98.71 |

## K  THE EFFECT OF VARYING $\lambda$

We investigate the effect of choosing the training hyperparameter $\lambda$, which is the factor of the respective loss on the out-of-distribution samples and represents $\frac{p(o)}{p(i)}$ during training. We evaluate models trained with Outlier Exposure Hendrycks et al. (2019a), background class and shared training of binary discriminator and classifier, all scored with $s_3$ (the implicit scoring function of OE). Note that in Section 4, the OE, BGC and SHARED models trained with $\lambda = 1.0$.

In Tables 12 and 13 we see that for CIFAR-10 the differences between different choices of $\lambda$ are minor with no clear favorite, but setting $\lambda = 2.0$ tends to be too high. For CIFAR-100, the differences are much larger. Here, choosing $\lambda$ too small can have a severe negative effect on the detection of the otherwise relatively to detect far out-distributions SVHN, Uniform Noise and Smooth Noise. Regarding the numbers for both in-distribution datasets, using $\lambda = 1$ is a considerate default choice.

Table 12: Effect of varying $\lambda$ during training for OE (Hendrycks et al., 2019a), the $s_3$ scoring function for models with background class and SHARED COMBI $s_3$. Shown are test accuracy and **FPR@95%TPR** with **OpenImages** as training out-distribution.

IN-DISTRIBUTION: CIFAR-10

| MODEL | ACC. | MEAN FPR | SVHN FPR | LSUN FPR | UNI FPR | SMOOTH FPR | C-100 FPR | 80M FPR | CELA FPR | OPENIM FPR |
|---|---|---|---|---|---|---|---|---|---|---|
| OE $\lambda$=0.1 | 95.36 | 14.34 | 13.31 | 0.00 | 0.25 | 0.03 | 48.37 | 36.65 | 1.77 | 12.60 |
| OE $\lambda$=0.25 | 95.31 | 16.17 | 19.64 | 0.00 | 0.00 | 0.05 | 51.77 | 40.21 | 1.55 | 7.74 |
| OE $\lambda$=0.5 | 95.27 | 15.98 | 16.23 | 0.00 | 0.07 | 0.01 | 53.06 | 41.72 | 0.79 | 4.85 |
| OE $\lambda$=1.0 | 95.06 | 15.20 | 9.58 | 0.00 | 0.00 | 0.00 | 54.05 | 42.33 | 0.45 | 3.46 |
| OE $\lambda$=2.0 | 95.19 | 15.98 | 12.93 | 0.00 | 0.00 | 0.00 | 55.99 | 42.68 | 0.26 | 1.67 |
| BGC $s_3$ $\lambda$=0.1 | 95.38 | 16.21 | 18.93 | 0.00 | 0.12 | 1.05 | 52.43 | 39.42 | 1.53 | 9.56 |
| BGC $s_3$ $\lambda$=0.25 | 95.27 | 15.83 | 12.76 | 0.00 | 0.04 | 0.03 | 54.06 | 43.26 | 0.66 | 5.43 |
| BGC $s_3$ $\lambda$=0.5 | 95.23 | 15.14 | 9.33 | 0.00 | 0.09 | 0.04 | 52.97 | 42.79 | 0.74 | 3.19 |
| BGC $s_3\lambda$=1.0 | 95.21 | 16.63 | 7.69 | 0.00 | 0.07 | 2.36 | 55.19 | 44.67 | 6.41 | 1.74 |
| BGC $s_3$ $\lambda$=2.0 | 95.28 | 17.45 | 8.72 | 0.00 | 0.00 | 0.00 | 59.02 | 48.09 | 6.31 | 0.71 |
| SC $s_3$ $\lambda$=0.1 | 95.20 | 16.49 | 15.22 | 0.00 | 0.27 | 0.00 | 54.30 | 43.97 | 1.69 | 10.46 |
| SC $s_3$ $\lambda$=0.25 | 95.21 | 16.17 | 15.32 | 0.00 | 0.15 | 0.04 | 54.60 | 42.59 | 0.53 | 5.19 |
| SC $s_3$ $\lambda$=0.5 | 95.25 | 16.02 | 9.21 | 0.00 | 0.09 | 0.00 | 55.72 | 45.55 | 1.56 | 2.75 |
| SC $s_3$ $\lambda$=1.0 | 95.28 | 16.06 | 9.00 | 0.00 | 0.00 | 0.00 | 58.68 | 42.85 | 1.91 | 0.66 |
| SC $s_3$ $\lambda$=2.0 | 95.26 | 17.04 | 9.33 | 0.00 | 0.00 | 0.00 | 58.20 | 45.81 | 5.92 | 0.68 |

IN-DISTRIBUTION: CIFAR-100

| MODEL | ACC. | MEAN FPR | SVHN FPR | LSUN FPR | UNI FPR | SMOOTH FPR | C-10 FPR | 80M FPR | OPENIM FPR |
|---|---|---|---|---|---|---|---|---|---|
| OE $\lambda$=0.1 | 77.28 | 52.35 | 70.29 | 1.00 | 28.58 | 56.15 | 82.44 | 75.64 | 17.94 |
| OE $\lambda$=0.25 | 76.96 | 46.40 | 60.70 | 0.00 | 9.24 | 48.89 | 83.01 | 76.59 | 6.01 |
| OE $\lambda$=0.5 | 77.22 | 36.43 | 53.12 | 0.00 | 0.15 | 4.62 | 82.86 | 77.84 | 4.23 |
| OE $\lambda$=1.0 | 77.19 | 35.03 | 47.36 | 0.00 | 0.67 | 0.08 | 84.64 | 77.42 | 1.28 |
| OE $\lambda$=2.0 | 76.95 | 33.25 | 37.04 | 0.00 | 0.00 | 1.70 | 83.28 | 77.48 | 0.86 |
| BGC $s_3$ $\lambda$=0.1 | 76.87 | 41.23 | 67.39 | 1.00 | 5.97 | 13.81 | 82.44 | 76.76 | 15.70 |
| BGC $s_3$ $\lambda$=0.25 | 77.05 | 42.05 | 49.94 | 0.00 | 17.07 | 25.41 | 82.34 | 77.55 | 7.47 |
| BGC $s_3$ $\lambda$=0.5 | 77.17 | 36.14 | 43.90 | 0.00 | 10.54 | 3.70 | 82.86 | 75.86 | 3.76 |
| BGC $s_3\lambda$=1.0 | 77.61 | 33.36 | 37.27 | 0.00 | 0.00 | 0.20 | 84.51 | 78.19 | 1.27 |
| BGC $s_3$ $\lambda$=2.0 | 77.26 | 32.60 | 35.34 | 0.00 | 0.00 | 0.00 | 82.69 | 77.57 | 1.20 |
| SC $s_3$ $\lambda$=0.1 | 77.35 | 58.54 | 62.07 | 0.33 | 92.62 | 37.44 | 82.05 | 76.74 | 14.17 |
| SC $s_3$ $\lambda$=0.25 | 76.96 | 41.34 | 63.29 | 0.00 | 10.24 | 14.34 | 82.68 | 77.50 | 7.22 |
| SC $s_3$ $\lambda$=0.5 | 76.64 | 49.63 | 49.55 | 0.00 | 44.78 | 43.85 | 82.30 | 77.31 | 1.97 |
| SC $s_3$ $\lambda$=1.0 | 77.35 | 33.06 | 37.57 | 0.00 | 0.00 | 1.13 | 82.68 | 77.01 | 1.85 |
| SC $s_3$ $\lambda$=2.0 | 76.63 | 34.09 | 39.56 | 0.00 | 0.00 | 5.21 | 82.57 | 77.18 | 1.04 |

Table 13: Effect of varying $\lambda$ during training for OE (Hendrycks et al., 2019a), the $s_3$ scoring function for models with background class and SHARED COMBI $s_3$. Shown are test accuracy and **AUC** (AUROC) with **OpenImages** as training out-distribution.

IN-DISTRIBUTION: CIFAR-10

| MODEL | ACC. | MEAN AUC | SVHN AUC | LSUN AUC | UNI AUC | SMOOTH AUC | C-100 AUC | 80M AUC | CELA AUC | OPENIM AUC |
|---|---|---|---|---|---|---|---|---|---|---|
| OE $\lambda$=0.1 | 95.36 | 97.49 | 97.96 | 99.96 | 99.84 | 99.93 | 91.25 | 93.81 | 99.70 | 97.86 |
| OE $\lambda$=0.25 | 95.31 | 97.15 | 96.93 | 99.98 | 99.98 | 99.93 | 90.59 | 92.91 | 99.74 | 98.71 |
| OE $\lambda$=0.5 | 95.27 | 97.19 | 97.53 | 99.99 | 99.95 | 99.99 | 90.29 | 92.73 | 99.87 | 99.20 |
| OE $\lambda$=1.0 | 95.06 | 97.28 | 98.49 | 99.99 | 99.99 | 99.99 | 90.03 | 92.53 | 99.91 | 99.43 |
| OE $\lambda$=2.0 | 95.19 | 97.19 | 97.99 | 99.99 | 99.99 | 100.00 | 89.89 | 92.56 | 99.93 | 99.67 |
| BGC $s_3$ $\lambda$=0.1 | 95.38 | 97.37 | 97.32 | 99.99 | 99.93 | 99.63 | 91.18 | 93.74 | 99.78 | 98.39 |
| BGC $s_3$ $\lambda$=0.25 | 95.27 | 97.43 | 98.25 | 100.00 | 99.96 | 99.98 | 90.85 | 93.09 | 99.91 | 99.13 |
| BGC $s_3$ $\lambda$=0.5 | 95.23 | 97.44 | 98.62 | 100.00 | 99.93 | 99.98 | 90.80 | 92.84 | 99.89 | 99.48 |
| BGC $s_3$ $\lambda$=1.0 | 95.21 | 97.21 | 98.87 | 100.00 | 99.98 | 99.62 | 90.47 | 92.41 | 99.08 | 99.71 |
| BGC $s_3$ $\lambda$=2.0 | 95.28 | 97.10 | 98.71 | 100.00 | 99.98 | 100.00 | 89.93 | 91.94 | 99.11 | 99.86 |
| SC $s_3$ $\lambda$=0.1 | 95.20 | 97.29 | 97.87 | 99.99 | 99.91 | 99.98 | 90.69 | 92.82 | 99.77 | 98.32 |
| SC $s_3$ $\lambda$=0.25 | 95.21 | 97.33 | 97.90 | 99.99 | 99.91 | 99.97 | 90.57 | 93.08 | 99.92 | 99.17 |
| SC $s_3$ $\lambda$=0.5 | 95.25 | 97.34 | 98.68 | 100.00 | 99.95 | 100.00 | 90.39 | 92.56 | 99.78 | 99.57 |
| SC $s_3$ $\lambda$=1.0 | 95.28 | 97.26 | 98.62 | 100.00 | 99.93 | 99.94 | 89.75 | 92.85 | 99.71 | 99.88 |
| SC $s_3$ $\lambda$=2.0 | 95.26 | 97.13 | 98.59 | 100.00 | 100.00 | 99.99 | 89.95 | 92.25 | 99.13 | 99.88 |

IN-DISTRIBUTION: CIFAR-100

| MODEL | ACC. | MEAN AUC | SVHN AUC | LSUN AUC | UNI AUC | SMOOTH AUC | C-10 AUC | 80M AUC | | OPENIM AUC |
|---|---|---|---|---|---|---|---|---|---|---|
| OE $\lambda$=0.1 | 77.28 | 87.38 | 83.69 | 99.79 | 94.79 | 91.50 | 76.08 | 78.43 | | 95.73 |
| OE $\lambda$=0.25 | 76.96 | 88.31 | 86.94 | 99.98 | 97.74 | 90.74 | 76.11 | 78.35 | | 98.51 |
| OE $\lambda$=0.5 | 77.22 | 89.82 | 87.22 | 99.98 | 99.74 | 98.95 | 75.39 | 77.64 | | 98.94 |
| OE $\lambda$=1.0 | 77.19 | 90.37 | 89.54 | 99.98 | 99.03 | 99.68 | 75.95 | 78.03 | | 99.67 |
| OE $\lambda$=2.0 | 76.95 | 90.56 | 91.42 | 99.99 | 99.78 | 99.08 | 75.51 | 77.58 | | 99.76 |
| BGC $s_3$ $\lambda$=0.1 | 76.87 | 88.86 | 83.41 | 99.87 | 98.35 | 97.37 | 75.97 | 78.21 | | 95.99 |
| BGC $s_3$ $\lambda$=0.25 | 77.05 | 89.29 | 89.10 | 99.99 | 97.13 | 95.91 | 75.96 | 77.68 | | 98.08 |
| BGC $s_3$ $\lambda$=0.5 | 77.17 | 90.17 | 90.07 | 99.99 | 97.99 | 99.07 | 75.49 | 78.42 | | 99.04 |
| BGC $s_3$ $\lambda$=1.0 | 77.61 | 90.46 | 90.46 | 99.99 | 99.88 | 99.74 | 74.88 | 77.82 | | 99.64 |
| BGC $s_3$ $\lambda$=2.0 | 77.26 | 90.83 | 91.74 | 99.99 | 99.72 | 99.80 | 75.72 | 78.02 | | 99.67 |
| SC $s_3$ $\lambda$=0.1 | 77.35 | 86.08 | 85.56 | 99.95 | 84.55 | 93.24 | 75.27 | 77.89 | | 96.74 |
| SC $s_3$ $\lambda$=0.25 | 76.96 | 88.76 | 85.08 | 99.98 | 97.73 | 97.17 | 75.11 | 77.51 | | 98.31 |
| SC $s_3$ $\lambda$=0.5 | 76.64 | 88.42 | 88.80 | 99.99 | 94.61 | 93.94 | 75.63 | 77.52 | | 99.51 |
| SC $s_3$ $\lambda$=1.0 | 77.35 | 90.73 | 91.69 | 99.99 | 99.57 | 99.53 | 75.50 | 78.10 | | 99.57 |
| SC $s_3$ $\lambda$=2.0 | 76.63 | 90.54 | 91.24 | 99.99 | 99.86 | 98.93 | 75.30 | 77.91 | | 99.74 |

## L  STATISTICS OVER FIVE RUNS WITH DIFFERENT SEEDS

Table 14: **Mean** $\mu$ and **standard deviation** $\sigma$ of the **FPR@95%TPR** measure for different methods and scoring functions over five runs each for models with **OpenImages** as training out-distribution. The training details are the same as for the results shown in Table 1.

IN-DISTRIBUTION: CIFAR-10

| MODEL | ACC. | MEAN FPR | SVHN FPR | LSUN FPR | UNI FPR | SMOOTH FPR | C-100 FPR | 80M FPR | CELA FPR | OPENIM FPR |
|---|---|---|---|---|---|---|---|---|---|---|
| OE $\mu$ | 95.11 | **15.49** | 11.42 | 0.00 | 0.00 | 0.09 | 54.35 | 42.14 | 0.40 | 2.75 |
| OE $\sigma$ | 0.04 | 0.32 | 1.75 | 0.00 | 0.00 | 0.18 | 0.61 | 0.55 | 0.11 | 0.49 |
| BGC $s_1$ $\mu$ | | 19.18 | 3.86 | 0.00 | 0.00 | 0.00 | 73.03 | 56.76 | 0.60 | 0.05 |
| BGC $s_1$ $\sigma$ | | 0.32 | 0.97 | 0.00 | 0.00 | 0.00 | 1.17 | 1.19 | 0.32 | 0.01 |
| BGC $s_2$ $\mu$ | **95.29** | 15.96 | 8.48 | 0.00 | 0.04 | 0.42 | 55.41 | 44.52 | 2.87 | 1.09 |
| BGC $s_2$ $\sigma$ | 0.10 | 0.63 | 0.88 | 0.00 | 0.05 | 0.84 | 1.57 | 1.55 | 1.94 | 0.34 |
| BGC $s_3$ $\mu$ | **95.29** | 15.99 | 8.69 | 0.00 | 0.04 | 0.47 | 55.29 | 44.49 | 2.94 | 1.16 |
| BGC $s_3$ $\sigma$ | 0.10 | 0.63 | 0.91 | 0.00 | 0.06 | 0.95 | 1.53 | 1.53 | 1.99 | 0.35 |
| SHARED BINDISC $\mu$ | | 20.45 | 5.30 | 0.00 | 0.00 | 0.00 | 77.19 | 59.63 | 1.06 | 0.05 |
| SHARED BINDISC $\sigma$ | | 0.49 | 1.02 | 0.00 | 0.00 | 0.00 | 1.09 | 3.00 | 0.46 | 0.02 |
| SHARED CLASSI $\mu$ | 95.21 | 33.21 | 26.37 | 9.33 | 49.73 | 11.31 | 57.34 | 48.37 | 29.98 | 36.73 |
| SHARED CLASSI $\sigma$ | 0.07 | 3.81 | 1.24 | 1.35 | 29.63 | 12.22 | 1.18 | 1.19 | 3.85 | 1.67 |
| SHARED COMBI $s_2$ $\mu$ | 95.21 | 16.28 | 8.77 | 0.00 | 0.20 | 0.00 | 57.14 | 44.92 | 2.92 | 0.95 |
| SHARED COMBI $s_2$ $\sigma$ | 0.07 | 0.39 | 1.11 | 0.00 | 0.38 | 0.00 | 1.39 | 1.37 | 1.42 | 0.23 |
| SHARED COMBI $s_3$ $\mu$ | 95.21 | 16.35 | 9.06 | 0.00 | 0.24 | 0.00 | 57.13 | 45.02 | 3.03 | 1.03 |
| SHARED COMBI $s_3$ $\sigma$ | 0.07 | 0.41 | 1.13 | 0.00 | 0.46 | 0.00 | 1.38 | 1.38 | 1.47 | 0.24 |

IN-DISTRIBUTION: CIFAR-100

| MODEL | ACC. | MEAN FPR | SVHN FPR | LSUN FPR | UNI FPR | SMOOTH FPR | C-10 FPR | 80M FPR | | OPENIM FPR |
|---|---|---|---|---|---|---|---|---|---|---|
| OE $\mu$ | 77.13 | 35.16 | 45.27 | 0.00 | 0.13 | 5.15 | 83.47 | 76.93 | | 1.66 |
| OE $\sigma$ | 0.23 | 0.80 | 5.35 | 0.00 | 0.27 | 5.35 | 0.89 | 0.59 | | 0.54 |
| BGC $s_1$ $\mu$ | | **31.12** | 11.88 | 0.00 | 0.00 | 0.12 | 93.85 | 80.85 | | 0.08 |
| BGC $s_1$ $\sigma$ | | 0.41 | 2.20 | 0.00 | 0.00 | 0.24 | 0.15 | 0.98 | | 0.03 |
| BGC $s_2$ $\mu$ | **77.30** | 35.54 | 40.11 | 0.00 | 7.58 | 4.90 | 83.44 | 77.20 | | 1.67 |
| BGC $s_2$ $\sigma$ | 0.35 | 2.12 | 5.67 | 0.00 | 10.26 | 5.35 | 0.80 | 0.73 | | 0.55 |
| BGC $s_3$ $\mu$ | **77.30** | 35.64 | 40.33 | 0.00 | 7.89 | 4.98 | 83.43 | 77.21 | | 1.69 |
| BGC $s_3$ $\sigma$ | 0.35 | 2.16 | 5.68 | 0.00 | 10.68 | 5.40 | 0.81 | 0.73 | | 0.56 |
| SHARED BINDISC $\mu$ | | 32.16 | 13.43 | 0.00 | 0.00 | 0.00 | 95.13 | 84.39 | | 0.05 |
| SHARED BINDISC $\sigma$ | | 0.22 | 2.10 | 0.00 | 0.00 | 0.00 | 0.19 | 0.83 | | 0.02 |
| SHARED CLASSI $\mu$ | 77.11 | 56.52 | 64.95 | 2.73 | 81.18 | 31.85 | 81.61 | 76.80 | | 22.97 |
| SHARED CLASSI $\sigma$ | 0.19 | 6.86 | 5.79 | 1.24 | 31.58 | 20.90 | 0.53 | 0.90 | | 2.95 |
| SHARED COMBI $s_2$ $\mu$ | 77.11 | 33.13 | 35.93 | 0.00 | 0.00 | 3.17 | 83.26 | 76.43 | | 1.13 |
| SHARED COMBI $s_2$ $\sigma$ | 0.19 | 0.59 | 4.68 | 0.00 | 0.00 | 4.35 | 0.51 | 0.82 | | 0.36 |
| SHARED COMBI $s_3$ $\mu$ | 77.11 | 33.18 | 36.17 | 0.00 | 0.00 | 3.23 | 83.24 | 76.43 | | 1.14 |
| SHARED COMBI $s_3$ $\sigma$ | 0.19 | 0.59 | 4.70 | 0.00 | 0.00 | 4.40 | 0.51 | 0.81 | | 0.38 |

Table 15: **Mean** $\mu$ and **standard deviation** $\sigma$ of the **AUROC** measure for different methods and scoring functions over five runs each for models with **OpenImages** as training out-distribution. The training details are the same as for the results shown in Table 5.

IN-DISTRIBUTION: CIFAR-10

| MODEL | ACC. | MEAN AUC | SVHN AUC | LSUN AUC | UNI AUC | SMOOTH AUC | C-100 AUC | 80M AUC | CELA AUC | OPENIM AUC |
|---|---|---|---|---|---|---|---|---|---|---|
| OE $\mu$ | 95.11 | 97.25 | 98.23 | 99.99 | 99.96 | 99.93 | 90.13 | 92.58 | 99.91 | 99.52 |
| OE $\sigma$ | 0.04 | 0.05 | 0.26 | 0.01 | 0.04 | 0.07 | 0.08 | 0.15 | 0.01 | 0.06 |
| BGC $s_1$ $\mu$ | | 94.59 | 99.16 | 100.00 | 99.98 | 99.97 | 78.02 | 85.15 | 99.85 | 99.96 |
| BGC $s_1$ $\sigma$ | | 0.27 | 0.21 | 0.00 | 0.02 | 0.02 | 1.02 | 0.85 | 0.07 | 0.01 |
| BGC $s_2$ $\mu$ | **95.29** | **97.33** | 98.74 | 100.00 | 99.98 | 99.92 | 90.48 | 92.58 | 99.59 | 99.81 |
| BGC $s_2$ $\sigma$ | 0.10 | 0.08 | 0.12 | 0.00 | 0.03 | 0.13 | 0.18 | 0.22 | 0.27 | 0.05 |
| BGC $s_3$ $\mu$ | **95.29** | 97.32 | 98.71 | 100.00 | 99.97 | 99.91 | 90.49 | 92.58 | 99.58 | 99.80 |
| BGC $s_3$ $\sigma$ | 0.10 | 0.08 | 0.12 | 0.00 | 0.04 | 0.15 | 0.18 | 0.22 | 0.28 | 0.05 |
| SHARED BINDISC $\mu$ | | 92.10 | 98.68 | 100.00 | 99.94 | 99.98 | 67.81 | 78.51 | 99.75 | 99.95 |
| SHARED BINDISC $\sigma$ | | 0.42 | 0.23 | 0.00 | 0.04 | 0.02 | 1.30 | 1.85 | 0.10 | 0.01 |
| SHARED CLASSI $\mu$ | 95.21 | 95.10 | 96.36 | 98.45 | 94.26 | 98.15 | 90.29 | 92.18 | 95.98 | 93.27 |
| SHARED CLASSI $\sigma$ | 0.07 | 0.39 | 0.18 | 0.15 | 3.07 | 1.21 | 0.18 | 0.21 | 0.46 | 0.34 |
| SHARED COMBI $s_2$ $\mu$ | 95.21 | 97.24 | 98.67 | 100.00 | 99.92 | 99.98 | 90.07 | 92.47 | 99.58 | 99.84 |
| SHARED COMBI $s_2$ $\sigma$ | 0.07 | 0.03 | 0.17 | 0.00 | 0.08 | 0.02 | 0.23 | 0.23 | 0.20 | 0.03 |
| SHARED COMBI $s_3$ $\mu$ | 95.21 | 97.24 | 98.64 | 100.00 | 99.91 | 99.98 | 90.09 | 92.48 | 99.56 | 99.83 |
| SHARED COMBI $s_3$ $\sigma$ | 0.07 | 0.04 | 0.17 | 0.00 | 0.10 | 0.02 | 0.22 | 0.23 | 0.20 | 0.03 |

IN-DISTRIBUTION: CIFAR-100

| MODEL | ACC. | MEAN AUC | SVHN AUC | LSUN AUC | UNI AUC | SMOOTH AUC | C-10 AUC | 80M AUC | OPENIM AUC |
|---|---|---|---|---|---|---|---|---|---|
| OE $\mu$ | 77.13 | 90.24 | 89.58 | 99.98 | 99.66 | 98.76 | 75.50 | 77.96 | 99.54 |
| OE $\sigma$ | 0.23 | 0.16 | 1.10 | 0.00 | 0.32 | 0.77 | 0.42 | 0.28 | 0.13 |
| BGC $s_1$ $\mu$ | | 88.48 | 97.17 | 99.99 | 99.61 | 99.57 | 60.96 | 73.57 | 99.92 |
| BGC $s_1$ $\sigma$ | | 0.17 | 0.38 | 0.00 | 0.14 | 0.21 | 0.51 | 0.38 | 0.01 |
| BGC $s_2$ $\mu$ | **77.30** | 90.23 | 90.45 | 99.99 | 98.60 | 98.90 | 75.34 | 78.12 | 99.55 |
| BGC $s_2$ $\sigma$ | 0.35 | 0.27 | 1.10 | 0.00 | 1.50 | 1.04 | 0.32 | 0.31 | 0.13 |
| BGC $s_3$ $\mu$ | **77.30** | 90.22 | 90.41 | 99.99 | 98.56 | 98.89 | 75.34 | 78.12 | 99.55 |
| BGC $s_3$ $\sigma$ | 0.35 | 0.28 | 1.10 | 0.00 | 1.55 | 1.05 | 0.32 | 0.31 | 0.13 |
| SHARED BINDISC $\mu$ | | 84.80 | 96.66 | 99.99 | 99.64 | 99.54 | 48.89 | 64.09 | 99.93 |
| SHARED BINDISC $\sigma$ | | 0.24 | 0.55 | 0.00 | 0.12 | 0.13 | 1.01 | 1.00 | 0.01 |
| SHARED CLASSI $\mu$ | 77.11 | 84.42 | 84.18 | 99.40 | 75.39 | 93.54 | 75.82 | 78.16 | 94.82 |
| SHARED CLASSI $\sigma$ | 0.19 | 2.29 | 2.09 | 0.20 | 12.57 | 5.07 | 0.14 | 0.25 | 0.68 |
| SHARED COMBI $s_2$ $\mu$ | 77.11 | **90.71** | 91.89 | 99.99 | 99.60 | 99.23 | 75.31 | 78.24 | 99.72 |
| SHARED COMBI $s_2$ $\sigma$ | 0.19 | 0.10 | 1.11 | 0.00 | 0.07 | 0.68 | 0.20 | 0.23 | 0.08 |
| SHARED COMBI $s_3$ $\mu$ | 77.11 | 90.70 | 91.85 | 99.99 | 99.59 | 99.22 | 75.32 | 78.24 | 99.72 |
| SHARED COMBI $s_3$ $\sigma$ | 0.19 | 0.10 | 1.11 | 0.00 | 0.08 | 0.69 | 0.20 | 0.23 | 0.08 |

