# OpenReview forum: "Revisiting Out-of-Distribution Detection: A Simple Baseline is Surprisingly Effective"
_ICLR.cc/2022/Conference — ICLR 2022 Submitted_

### Official Review · Reviewer_YRfA · 2021-10-29

**Correctness:** 4
**Technical Novelty And Significance:** 3
**Empirical Novelty And Significance:** 3
**Recommendation:** 8
**Confidence:** 2

**Main Review:**

Strengths:
* The paper is well-written and relatively easy to follow.
* All three of the author's contributions seem interesting and well-supported.
* I feel that the authors achieve notable progress towards their stated goal (a "better understanding of the key components of different OOD detection methods and to identify the key properties which lead to SOTA OOD detection performance"), particularly with their first and third contributions.

Weaknesses:
* In my view, the authors do not make particularly clear what the problem their work is aiming to solve is or why it needs to be solved. What's missing from the existing literature, and why is it important to fill in that gap? Their entire introduction is essentially a "related work" section, and then suddenly shifts to a list of three contributions. But the introduction does not explain the need for these contributions. Their stated goal (which I quoted above) comes across as quite vague. It is therefore also difficult for me as a reviewer (with limited expertise in this area) to make a judgement call as to the importance of this paper.

**Summary Of The Paper:**

The authors aim to to recognize common objectives in OOD detection as well as to identify the implicit scoring functions of different OOD detection methods. They show that binary discrimination between in- and (different) out distributions is equivalent to several different formulations of the OOD detection problem. They find that, when trained in a shared fashion with a standard classifier, this binary
discriminator reaches an OOD detection performance similar to that of Outlier Exposure.

**Summary Of The Review:**

The authors present well-supported contributions that seem interesting, but do little to explain what problem their contributions solve or why their contributions are needed.

---

> ### Author Response · Authors · 2021-11-22
> **Response to review by YRfA**
>
> ### We thank the reviewer for their time and the helpful comments.
>
> **"In my view, the authors do not make particularly clear what the problem their work is aiming to solve is or why it needs to be solved. What's missing from the existing literature, and why is it important to fill in that gap? Their entire introduction is essentially a "related work" section, and then suddenly shifts to a list of three contributions. But the introduction does not explain the need for these contributions. Their stated goal (which I quoted above) comes across as quite vague. "** \
> The community has so far focused on establishing novel OOD detection methods which are often empirically driven. We think it is important to also have an understanding of the foundations of the methods, in particular the implicit decision criterion which the OOD detection methods are using. This is important to identify differences but also to show that different methods effectively use the same criterion but just differ in the way this criterion is estimated. We think it is important to distinguish the estimation process of a criterion from the criterion itself.
>
> We definitely agree that in the introduction we jumped too fast from the related work to the contributions of the paper. Therefore we now have rewritten the introduction and hopefully better motivate why the contributions of this paper will lead to a better understanding of the differences and similarities between different OOD detection methods. Following a suggestion of Reviewer iH61, we have added the analysis of another quite different method with Liu et al.'s "Energy-based Out-of-distribution Detection" (NeurIPS 2020) in Appendix F, which shows that a quite rich class of methods can be analyzed with our techniques.

---

### Official Review · Reviewer_Lwwq · 2021-11-01

**Correctness:** 4
**Technical Novelty And Significance:** 4
**Empirical Novelty And Significance:** Not applicable
**Recommendation:** 10
**Confidence:** 3

**Main Review:**

+ (+) The paper is clearly written, has a good story line, is self contained and appendix A is useful/sufficient to grasp all mathematical parts.
+ (+) The main contribution of the paper is to unify under the same mathematical language different methods that were previously related less clearly/effectively.
+ (+) In the theoretical part of the paper - especially Sec. 3.1 and 3.2 -, mathematical steps are always interwined with intuitive but sound explanations (that I personally found eye-opening at times).

+ (-) The analysis framework is based on bayes optimality which is asymptotic - as the authors state in the ethics and reproducibility statement section at the end of the manuscript. But this is not an ethic nor a reproducibility matter: it is a peculiarity / design choice of this work that should be crystal clear to the reader, who will decide if this is a limitation she cares about or less. I would have liked to read it in Sec. 2 or Sec. 3 at most.

**Summary Of The Paper:**

In this paper, the authors bring together recent work in the OOD detection problem and provide to the reader a sound mathematical framework to understand similarities and differences among these. The framework is based on the equivalence class of scoring function under the AUC / FPR@qTPR metrics and bayes optimality. The tools introduced in the paper allow the authors to explain why different methods perform largely similarly, when one scoring function should be preferred to others and draw conclusions regarding training with / without label data of the in-distribution set.

**Summary Of The Review:**

Theoretical insights that unify different methods under the same light and allow new analysis are so valuable to the community, especially when they are so clearly exposed. The authors also show these theoretical insights are actionable, by drawing new conclusions later supported by empirical evidence. I think people working in the field will benefit from such a clear and encompassing narrative.

My review could be positively biased: Since I am not an active researcher in the field it was difficult for me to provide absolute meaning to the numbers in the experimental section, although on a relative scale I was able to follow the authors' conclusions. Because of my lack of expertise in this specific field I cannot provide positive nor negative feedback on this and on the related work reporting.

For these reasons I will take into great consideration all other reviews and discussion for the final score.

---

> ### Author Response · Authors · 2021-11-22
> **Response to review by Lwwq**
>
> ### We thank the reviewer for their time and the encouraging and helpful comments.
>
> **"The analysis framework is based on bayes optimality which is asymptotic - as the authors state in the ethics and reproducibility statement section at the end of the manuscript. But this is not an ethic nor a reproducibility matter: it is a peculiarity / design choice of this work that should be crystal clear to the reader, who will decide if this is a limitation she cares about or less. I would have liked to read it in Sec. 2 or Sec. 3 at most."** \
> We agree that the technical limitations should be mentioned in the discussion of the methods rather than in the Ethics and Reproducibility statement. We have moved their discussion to the beginning of Section 3 in the updated paper.

---

### Official Review · Reviewer_iH61 · 2021-11-02

**Correctness:** 3
**Technical Novelty And Significance:** 2
**Empirical Novelty And Significance:** 2
**Recommendation:** 6
**Confidence:** 4

**Main Review:**

This uses new dataset i.e. OpenImages as $D^{ood}_{train}$ instead of TinyImages because it was retracted for ethical concerns. How was this decision to choose this specific dataset motivated? Are there any specific characteristics that you searched for? Have you explored how this train OOD dataset should be chosen?
The method addresses many OOD methods in the literature from the past, but refrained from including other/more recent ones that uses self supervised[5][6], contrastive learning[7][8][9], adversarial resampling[11], energy based methods[9] etc. It would have been great if authors could discuss more on this and present how their method/derivation fits on any(not all) of these more recent frameworks/methods.

## Minor Issue:
I think there is a typo in lower table 2 heading, Did you mean C-10 instead of C-100 between SMOOTH and 80M FPR?

## Strength
This paper provides insights and findings of properties of different OOD detection approaches. This will help in getting more understanding of OOD methods and building novel ones further.  Most of the previous work were mainly empirical and lacked mathematical foundation
The finding of uncertainty based methods being equivalent under the hood with binary discriminators is interesting. It explains and provides some groundings to why those different methods might be performing similar ways.
It includes well-motivated experiments, their clean code implementation and results as well as a comprehensive appendix to support the claims.
## Weakness
The empirical significance and novelty of third contribution seems limited to me since the claim resembles the ones made by Thulasidasan et al. [2021] and Mohseni et al. [2020]. I’d request the authors to discuss more on the effect of their advancements(using unlabelled data($x_{r}^{IN}$) with BGC) and empirically discuss/show the significance of the gains.




**Summary Of The Paper:**

The paper analyzes different OOD detection methods and show that even if the formulation for many OOD methods were different, the binary discrimination is equivalent to those different types of methods when the rankings induced by Bayes optimal classifier are analyzed. They also derive implicit scoring functions for confidence loss of OE and for BGC and compare them with “optimal” scoring functions. They also claim that training binary discriminator (in-dist vs out-dist) in a shared fashion along with standard classifier reaches state-of-the-art OOD performance.

**Summary Of The Review:**

The paper provides a good survey and insights of existing OOD approaches discussing their similarities and differences. While I commend the mathematical derivation  and analysis of multiple existing OOD methods, the novelty of empirical contributions seem marginally significant to me. I am open to increasing my score  if authors can address some of my concerns above, provide some judgments on more recent methods and/or further insights on the statistical significance (like standard deviation) of the top performing models i.e.  OE, BGC and SHARED_COMBI and provide evidence to significance of the  superiority of SHARED_COMBI over others. Especially since the scores in tables for OE and BGC seem slightly different from the ones reported in the original paper (comparing Appendix F since a different dataset was used in main table).

### References:
[1] Neal, Lawrence, et al. "Open set learning with counterfactual images." in  ECCV 2018.\
[2] Thulasidasan, Sushil, et. al. (2021) "An Effective Baseline for Robustness to Distributional Shift"\
[3] Mohseni, et al. "Self-Supervised Learning for Generalizable Out-of-Distribution Detection." AAAI. 2020.\
[4] Zhang et. al. "Universum prescription: Regularization using unlabeled data." in AAAI 2017.\
[5] Hendrycks et. al. Using self-supervised learning can improve model robustness and uncertainty. In Neurips 2019.\
[6] Golan et. al. "Deep anomaly detection using geometric transformations." in Neurips 2018.\
[7] Winkens, Bunel, Roy et. al. (2020). Contrastive training for improved out-of-distribution detection.\
[8] Tack, Mo et. al "CSI: Novelty Detection via Contrastive Learning on Distributionally Shifted Instances" in Neurips 2020.\
[9] Liu et. al (2020) "Hybrid discriminative-generative training via contrastive learning."\
[10] Liu et al. "Energy-based Out-of-distribution Detection, in Neurips 2020."\
[11] Li et. al. "Background Data Resampling for Outlier-Aware Classification", in CVPR 2020.\
[12] Hendrycks et al. "Deep anomaly detection with outlier exposure". In ICLR, 2019.

---

> ### Author Response · Authors · 2021-11-22
> **Response to review by iH61 -- Part 1**
>
> #### We thank the reviewer for their time and the helpful comments.
>
> **"How was this decision to choose [specifically OpenImages] motivated? Are there any specific characteristics that you searched for? Have you explored how this train OOD dataset should be chosen?"** \
> We were guided by our own experience and the discussion of the authors of [OE] (Section 5): $D^{ood}_{train}$ should consist mainly of natural images, should have a high semantic diversity, and the number of samples in the dataset should be large. OpenImages fulfills the high semantic diversity having images from about 20.000 classes, mainly natural images, has about 9 million training samples and does not contain ethically problematic images. We initially also did checks if some artifacts were introduced by the downscaling which would make it possible for a neural network to separate perfectly in-distribution from OpenImages by such a spurious feature, which was not the case.
>
> It is, however, to be noted that CIFAR was sourced as a subset of 80M Tiny Images, see [CIFAR] webpage, which explains the somewhat better results with 80M as $D^{ood}_{train}$ that we observe in Appendix F (images have gone through the same image preprocessing pipeline).
> We think that in certain practical situations, availability of a large general image dataset that one has not been able to obtain by the same data collection process as for the in-distribution is indeed a more realistic scenario.
>
>
> **"The method addresses many OOD methods in the literature from the past, but refrained from including other/more recent ones that uses self supervised[5][6], contrastive learning[7][8][9], adversarial resampling[11], energy based methods[9] etc. It would have been great if authors could discuss more on this and present how their method/derivation fits on any(not all) of these more recent frameworks/methods."** \
> We are thankful for pointing out these interesting further OOD detection methods. In our related work section which is now in Appendix D, we have included short discussions on the relation of these methods to those we analyze in our paper.
> While in this short amount of time, a full check of all these methods is infeasible, we managed to do an analysis of  Liu et al.'s "Energy-based Out-of-distribution Detection" (NeurIPS 2020) which use the energy of a jointly trained classifier as score function. In Appendix F we prove that the Bayes optimal energy is equivalent to the binary disriminator between in- and out-distribution. This shows that our approach generalizes to quite distinct methods and further stresses the similarity of existing approaches.
>
> **"I think there is a typo in lower table 2 heading, Did you mean C-10 instead of C-100 between SMOOTH and 80M FPR?"** \
> Thanks for making us aware of the typo, the numbers shown in this column were indeed obtained with CIFAR-10 as the test out-distribution.
>
> [OE] Hendrycks et al.: "Deep anomaly detection with outlier exposure". ICLR, 2019. \
> [CIFAR] Alex Krizhevsky (2009): Learning Multiple Layers of Features from Tiny Images, https://www.cs.toronto.edu/~kriz/cifar.html

---

> ### Author Response · Authors · 2021-11-22
> **Response to review by iH61 -- Part 2**
>
> **"The empirical significance and novelty of third contribution seems limited to me since the claim resembles the ones made by Thulasidasan et al. [2021] and Mohseni et al. [2020]. I’d request the authors to discuss more on the effect of their advancements(using unlabelled data $x_r^{IN}$ with BGC) and empirically discuss/show the significance of the gains."** \
> We would like to stress that Thulasidasan et al (2021) and Mohseni et al. (2020) are aiming at good empirical OOD detection and do not identify relations between different methods as it is the goal of our paper.
> The DAC score of Thulasidasan et al. is indeed what we include as BGC training in our analysis, and Mohseni et al.'s approach of splitting up the OOD output into multiple self-learned classes is very similar.
> Note, however, that the main point of our analysis of BGC models is how they can be used to model different OOD detection approaches like OE and a dedicated binary discriminator.
> Our third contribution is that we show that a predictor that purely discriminates between in- and training-out-distribution can indeed be directly used for OOD detection (if its training is supported by weight sharing with a classifier).
>
> **"further insights on the statistical significance (like standard deviation)  of the top performing models i.e. OE, BGC and SHARED_COMBI and provide evidence to significance of the superiority of SHARED_COMBI over others."** \
> In the updated version, we have included mean and standard deviation over five runs of the different methods and scoring functions in Appendix L.
> The standard deviations are quite small in almost all cases.
> The conclusions are unchanged.
>
> **"the scores in tables for OE and BGC seem slightly different from the ones reported in the original paper"** \
> The reason for the differences to the results reported in the OE paper is that they show evaluations of models that were only fine-tuned with the OE loss for only 10 epochs. They note in their Appendix C that training from scratch while more expensive gives superior results.
> Since we find the difference quite significant, and we are interested in models that are fully trained with their respective losses, we trained all of our models from scratch.
>
> On their github repo, they have evaluation files for both approaches:
>
> Fine-tuning (their Table A): https://github.com/hendrycks/outlier-exposure/blob/master/CIFAR/snapshots/cifar10_wrn_oe_tune_test.txt \
> From scratch (we evaluate these models in App. H (was F)): https://github.com/hendrycks/outlier-exposure/blob/master/CIFAR/snapshots/cifar10_wrn_oe_scratch_test.txt

---

> ### Comment · Reviewer_iH61 · 2021-11-27
> **Rebuttal response**
>
> I thank the authors for the elaborative response to my concerns and really delighted to see you were able to related your paper with a more recent approach in such a short time.
> I am happy to update the score to an “accept” now.
>
> However, I am still unable to see the novelty of third contribution showing binary discriminator when trained in a shared manner reaches SOTA since I think the previous  paper[esp. Thulasidasan et. al] already showed that. I think this is also seen in newly added appendix L that shows BGC performs consistently better than other methods including ‘Shared’ method proposed in this paper.
>
> For this reason, I am unable to give this paper a full/better score. The acceptance score is mainly for the framework and conclusions derived for different methods.

---

> > ### Author Response · Authors · 2021-11-29
> > **Re: Rebuttal response**
> >
> > We much appreciate the increased score and would like to resolve the remaining concern.
> >
> > As also mentioned in the discussion with Reviewer YhZ7, we agree that the third contribution indeed needs to be revised.
> > We would like to still emphasize in the contributions the important observation that a straightforward binary discriminator can be a useful tool for OOD detection, and we suggest replacing the third contribution with:
> >
> > * "We show that the combination of a binary discriminator between in- and out-distribution with a standard classifier on the in-distribution, when trained together in a shared fashion, yields OOD detection performance that is competitive with state-of-the-art methods."
> >
> > This is now well supported by the empirical results, as the mean FPR of Shared Combi $s_2$ is better in individual comparison with every other method in Table 1 (or 14) for at least one of the two in-distributions CIFAR-10 and CIFAR-100. The same holds for 80M (Table 7) and SVHN (Table 9) as training OOD, and further Shared Combi $s_2$ has the best mean FPR for Restricted ImageNet (Table 11).
> >
> > To elaborate concretely on the comparison mentioned in your comment, while in Appendix L the shared binary discriminator indeed falls behind BGC $s_1$ (i.e. the method of Thulasidasan et. al), note that Shared Combi $s_2$ (and $s_3$) performs significantly better than BGC $s_1$ with CIFAR-10 as in-distribution, so neither is consistently better or worse than the other.
> > This is consistent with our message that the different methods are closely related and that their relative performance can vary between applications.

---

### Official Review · Reviewer_YhZ7 · 2021-11-02

**Correctness:** 2
**Technical Novelty And Significance:** 2
**Empirical Novelty And Significance:** 2
**Recommendation:** 3
**Confidence:** 2

**Main Review:**

Strengths:

The paper is trying to demonstrate that a simple baseline can out-perform other more complicated methods. I appreciate such efforts and I think they provide a good check for the field in general.

Weaknesses:

1) The paper is poorly written and hard to follow. For example, the intro discusses why OOD detection is an important problem and then delves into related work, but fails to properly motivate the contributions by, say, discussing why the rankings induced by the Bayes optimal classifier are relevant, why we may want to revisit and better understand the differences between various OOD methods, or why we may want to consider a binary discriminator method. As another example, Section 2 presents theories and definitions related to when scoring functions are equivalent without explaining to the reader why it is necessary to understand or follow the math.

2) Table 1 actually shows that Outlier Exposure outperforms a Binary Classifier on average on the CIFAR-10 dataset? So only on one of the two datasets do we see the claimed result?

3) Experiments are not that thorough. Only CIFAR-10 and CIFAR-100 are used as in-distribution datasets, and OpenImages is the only out-of-distribution dataset.

3) A binary discriminator requires knowing which data is OOD at training time, and having a large amount of this OOD data. In practice, we may not have access to samples from the OOD dataset to train on, or we may have only a small number of samples. It's not clear to me whether other OOD detection methods avoid this requirement.  I would appreciate if there was more discussion about this topic.

**Summary Of The Paper:**

The authors show that 1) a simple baseline,  binary discriminator between in- and out-of- distribution data, is competitive with state-of-the-art OOD detection techniques and 2) the Bayes optimal scoring functions of several proposed methods for out-of-distribution detection are equivalent to the scoring functions of simple binary discriminators. The authors also aim to provide a better understanding of key components of various OOD detection methods such as Outlier Exposure.


**Summary Of The Review:**

The paper was poorly written and difficult to follow. I recommend that the authors try to make their presentation more clear and concise.

---

> ### Author Response · Authors · 2021-11-22
> **Response to review by YhZ7 -- Part 1**
>
> ### We thank the reviewer for their time and the helpful comments.
>
> **"The paper is trying to demonstrate that a simple baseline can out-perform other more complicated methods."** \
> We would like to emphasize that the goal of our paper is to show how different methods are related to each other, rather than establishing the superiority of any specific approach.
>
> **"The paper is poorly written and hard to follow. For example, the intro discusses why OOD detection is an important problem and then delves into related work, but fails to properly motivate the contributions by, say, discussing why the rankings induced by the Bayes optimal classifier are relevant, why we may want to revisit and better understand the differences between various OOD methods, or why we may want to consider a binary discriminator method. As another example, Section 2 presents theories and definitions related to when scoring functions are equivalent without explaining to the reader why it is necessary to understand or follow the math."** \
> The community has so far focused on establishing novel OOD detection methods which are often empirically driven. We think it is important to also have an understanding of the foundations of the methods, in particular the implicit decision criterion which the OOD detection methods are using. This is important to identify differences but also to show that different methods effectively use the same criterion but just differ in the way this criterion is estimated. We think it is important to distinguish the estimation process of a criterion from the criterion itself.
>
> We definitely agree that in the introduction we jumped too fast from the related work to the contributions of the paper. Therefore we now have rewritten the introduction and hopefully better motivate why the contributions of this paper will lead to a better understanding of the differences and similarities between different OOD detection methods. Following a suggestion of Reviewer iH61, we have added the analysis of another quite different method with Liu et al.'s "Energy-based Out-of-distribution Detection" (NeurIPS 2020) in Appendix F, which shows that a quite rich class of methods can be analyzed with our techniques.
>
> Regarding Section 2 - as a paper which wants to establish theoretical foundations and guarantees, we need to provide formal definitions first. The main point of this section is to characterize the set of transformations of a scoring function which leaves the OOD detection criteria like AUC or FPR invariant. This is important for the analysis later on, since the scoring functions of different methods are in many cases not identical but equivalent in the sense of Definition 1.
>
>
> **"Table 1 actually shows that Outlier Exposure outperforms a Binary Classifier on average on the CIFAR-10 dataset? So only on one of the two datasets do we see the claimed result?"** \
> We agree that there is no clear-cut 'winner' among the considered methods and scoring functions.
> Following the theoretical considerations, Outlier Exposure, BGC $s_3$ and Shared $s_3$ are expected to behave similarly, and different relative rankings between them for different datasets are a result of differences in the training dynamics.
> In this paper, our goal is to bring structure into the space of possible approaches given the assumptions on available training data. It is not the goal to show general superiority of certain methods, which would stand in contrast to the multiple theoretical equivalences between the models and their scoring functions which we show in the paper.
>
> **"Experiments are not that thorough. Only CIFAR-10 and CIFAR-100 are used as in-distribution datasets, and OpenImages is the only out-of-distribution dataset."** \
> We friendly disagree. We evaluate on two in-distribution tasks CIFAR-10 and CIFAR-100 and evaluate on seven resp. six out-of-distribution test sets. As training out-distribution we once use Open Images (main paper) and the retracted 80 Million Tiny Images dataset used in previous works (Appendix H). Moreover, we test SVHN as a training out-distribution to illustrate why a far-away training out-distribution fails to yield meaningful results as the support of in- and training out-distributions are disjoint (Appendix I).
>
> We now also added Restricted ImageNet as in-distribution dataset and the remaining ILSVRC2012 classes as training out-of-distribution dataset.
> The evaluations on Retricted ImageNet confirm the findings of the main paper and are discussed in Appendix J.
>
> [OE]: Hendrycks et al. "Deep anomaly detection with outlier exposure". In ICLR, 2019.

---

> > ### Comment · Reviewer_YhZ7 · 2021-11-29
> > **Response**
> >
> > *Contribution: binary discriminator reaches SOTA OOD detection performance*
> > The authors state that one of the main contributions of this paper is "We show that when training the binary discriminator between in- and out-distribution together with a standard classifier on the in-distribution in a shared fashion, the binary discriminator reaches state-of-the-art OOD detection performance."
> >
> > However in their response the authors say "We would like to emphasize that the goal of our paper is to show how different methods are related to each other, rather than establishing the superiority of any specific approach."
> >
> > These two statements seem contradictory to me and I feel that the contribution as stated is not supported by the empirical results.
> >
> >
> > - The intro is improved. Thank you for updating.
> > -The motivation you give for Section 2 is important and should be discussed in the beginning of the section so the reader understands why the theory is being introduced.
> > - Thank you for clarifying the different distributions you use in training and evaluation and for adding the Restricted ImageNet results.

---

> > > ### Author Response · Authors · 2021-11-29
> > > **Re: Response**
> > >
> > > We agree that the third contribution needs to be revised and that the evaluations allow no clear preference over all datasets for one of the different methods and scoring functions.
> > > We would like to still emphasize in the contributions the important observation that a straightforward binary discrimininator can be a useful tool for OOD detection, and we suggest replacing the third contribution with:
> > >
> > > * "We show that the combination of a binary discriminator between in- and out-distribution with a standard classifier on the in-distribution, when trained together in a shared fashion, yields OOD detection performance that is competitive with state-of-the-art methods."
> > >
> > > This is now well supported by the empirical results, as the mean FPR of Shared Combi $s_2$ is better in individual comparison with every other method in Table 1 (or 14) for at least one of the two in-distributions CIFAR-10 and CIFAR-100. The same holds for 80M (Table 7) and SVHN (Table 9) as training OOD, and further Shared Combi $s_2$ has the best mean FPR for Restricted ImageNet (Table 11).
> > >
> > > We are happy to include the motivations for the developed theory in the beginning of Section 2.
> > >
> > > After many of the original points have been resolved, we would much appreciate it if you would reconsider your score.

---

> ### Author Response · Authors · 2021-11-22
> **Response to review by YhZ7 -- Part 2**
>
> **"A binary discriminator requires knowing which data is OOD at training time, and having a large amount of this OOD data. In practice, we may not have access to samples from the OOD dataset to train on, or we may have only a small number of samples. It's not clear to me whether other OOD detection methods avoid this requirement. I would appreciate if there was more discussion about this topic."** \
> It is important to distinguish between three qualitatively different situations regarding availability of OOD data at training time: A) only in-distribution data available, B) access to in-distribution data and OOD data that is not related to test out-distributions , C) access to samples from the out-distributions that are encountered at test time.
>
> Several OOD detection methods that work under premise A, i.e. avoid the requirement of any non-in-distribution training data, have been proposed, but up to our knowledge, none of them achieve satisfactory OOD detection performance, particularly for close out-of-distribution data.
> We theoretically show towards the end of 3.1 that density estimators, which work with in-distribution training data only, implicitly presume uniform noise as training out-distribution, and we use this to explain their poor OOD detection performance.
> In Appendix E we analyze some additional methods that work under this premise A and observe that each has severe weaknesses on some test OOD datasets.
>
> The focus of the experiments in our paper is on understanding methods that work under premise B, which has been proposed in [OE]. This general situation is arguably very realistic for most practical applications. Particularly in any image recognition application, one has the application specific in-distribution training data, and one can always use one of the large scale image datasets like OpenImages as training OOD dataset.
> Since methods like OE are successful for multiple known tasks, it is reasonable to employ such methods for a given potentially OOD sensitive application.
> We provide an understanding of these methods and their variants.
> Note that our analysis does not depend on the quality in terms of distribution or size of the training OOD dataset; for suitable ones like OpenImages or 80 Million Tiny Images (Appendix H), the theoretically equivalent approaches behave similarly well, and for unsuited out-distributions like SVHN (Appendix I), they all fall behind even confidences of a plain classifier in several test cases.
>
> Finally, there are approaches that work under premise C and use the train set of the test out-*distribution* at training time (including fine-tuning).
> This is a markedly different problem setting, as the task becomes much easier in principle when it is not demanded that the model generalize to *unseen distributions*.
> We do not consider such settings as OOD detection in the sense of this paper.

---

> > ### Comment · Reviewer_YhZ7 · 2021-11-29
> > **Data assumptions**
> >
> > I appreciate this discussion. To strengthen the paper, I would include a discussion of the different data assumptions possible for OOD detection and describe why scenario B is most realistic.

---

> > > ### Author Response · Authors · 2021-11-29
> > > **Re: Data assumptions**
> > >
> > > We will gladly follow this suggestion and include a discussion of the different possible assumptions on the data available during training in the final version.

---

### Official Review · Reviewer_vYWv · 2021-11-03

**Correctness:** 3
**Technical Novelty And Significance:** 3
**Empirical Novelty And Significance:** 2
**Recommendation:** 6
**Confidence:** 4

**Main Review:**

The proposed method is not completely new, but the purpose of this paper is to identify common objectives as well as the identification of the implicit scoring functions of different OOD detection methods. Also the proposed method is simple and efficient. The connection to the previous work is comprehensively discussed.

I found the table results are a bit too busy to follow, with many different methods and scores. It would be great if you can highlight the best values in each column. Also, the FPR@95%TPR can be very sensitive. I would recommend to report AUROC in the main text.

Here is a relevant paper to the idea of training the binary discriminator between in- and out-distribution together with a standard classifier on the in-distribution in a shared fashion. It may be worth noting that.

Roy, Abhijit Guha, et al. "Does Your Dermatology Classifier Know What It Doesn't Know? Detecting the Long-Tail of Unseen Conditions." arXiv preprint arXiv:2104.03829 (2021).
https://arxiv.org/abs/2104.03829

**Summary Of The Paper:**

The paper shows that binary discrimination between in- and out- of distributions is equivalent to several generative model based OOD detection approaches such as likelihood ratios. Moreover, the paper shows that when the binary classifier between in- and out- trained in a shared fashion with a standard classifier for in-distribution classes, and using a score which integrates information from p(i|x) and p(y|x, i), the OOD detection performance achieves the state-of-the-art.

**Summary Of The Review:**

The paper provides helpful insights to connect methods for OOD detection tasks. It has both comprehensive theoretical and empirical analysis. The paper can be further polished to make it easy to follow.

---

> ### Author Response · Authors · 2021-11-22
> **Response to review by vYWv**
>
>
> ### We thank the reviewer for their time and the helpful comments.
>
> **"I found the table results are a bit too busy to follow, with many different methods and scores. It would be great if you can highlight the best values in each column."** \
> We agree that the tables are bit hard to parse, and we now highlight the best accuracy and the best mean FPR in bold. We suggest to highlight the best OOD result of each column differently by using dark green, as a good OOD detection method should provide consistently and simultaneously good results on all OOD test datasets.
>
> **"Also, the FPR@95%TPR can be very sensitive. I would recommend to report AUROC in the main text."** \
> Please note that we already report the AUC in Appendix G, but we are happy to move the AUC to the main part instead if this is consensus among the reviewers and the AC. We got opposite comments that we should show FPR in other papers. We think both together yield a good picture which is why we report both. We had to move one to the Appendix for space reasons.
>
> **"[[DermaOOD]]  is a relevant paper to the idea of training the binary discriminator between in- and out-distribution together with a standard classifier on the in-distribution in a shared fashion. It may be worth noting that."** \
> This paper is very interesting, especially since it also presents a valuable use case of OOD detection in a medical application.
> We have added a short discussion of their problem setting and method in the updated related works section in Appendix D.
> The premise of their method is slightly different from ours as they require class labels for the training out-distributions, which are not available to the models compared in our paper.
> Since evaluating the methods on the skin diseases dataset would be interesting, we had contacted the authors of [DermaDS] who contributed the dataset used in [DermaOOD], but due to legal reasons the teledermatology service with whom they collected the data does not provide public access to the data.
>
>
> [DermaDS] Liu et al.: "A deep learning system for differential diagnosis of skin diseases". In: Nat Med 26. \
> [DermaOOD] Roy et al.: "Does Your Dermatology Classifier Know What It Doesn't Know? Detecting the Long-Tail of Unseen Conditions". In:  Medical Imaging Analysis (2021).

---

### Author Response · Authors · 2021-11-22
**Changes in the updated paper**

We thank all reviewers for their insightful assessments and suggestions.
We have addressed all points in detail in the responses below the individual reviews.
In the updated paper, we have included and highlighted the following changes and additions:
 * Experiments with the new in-distribution **Restricted ImageNet** in Appendix J.
 * An analysis of  **Energy Based OOD Detection** [1] in Appendix F which proves the equivalence of the Bayes optimal energy to the binary discriminator between in-and out-distribution, with an experimental comparison in Appendix H.
 * More focus on the **motivations** behind our paper in the Introduction while the extended **Related Work** section has been moved to Appendix D for space reasons. If the reviewers and/or the AC prefer that we have the related work section in the main part, we can also try to put other parts of the paper into the appendix.
 * Further practical discussion on the **choice of the training OOD dataset** in Appendix H.
 * **Statistical evaluation** of mean and standard deviation for various methods in Appendix L.
 * The best results in each table column have been highlighted.


[1] Liu, Xiaoyun Wang, John Owens, and Yixuan Li. "Energy-based out-of-distribution detection." NeurIPS, 2020.

---

### Decision · Program_Chairs · 2022-01-20

**Decision:**

Reject

**Comment:**

The paper contributes to the understanding of out-of-distribution detection by showing that binary discrimination between in- and out-distribution examples 'is equivalent to several different formulations of the out-of-distribution detection problem'. The paper shows this in an asymptotic setup based on studying likelihood ratios for distinguishing in-distribution examples from out-of-distribution examples. The paper also provides numerical results showing that a simple baseline based on binary classification works well.

The paper got very mixed responses ranging from strong accept to reject:
- Reviewer YhZ7 (recommending 3: reject) raises several important concerns, specifically that the paper doesn't explain the significance of its contributions adequately, that experiments are not thorough enough (for example that only one out-of-distribution dataset is considered), and that to train a binary classifier one needs to have sufficiently many out-of-distribution examples.
The authors argued in response that the purpose of the paper is to provide an understanding of existing methods that are often empirically driven, made revisions to the exposition, and point out that they actually evaluate on six/seven out-of-distribution test sets.
After discussion, the reviewer is still concerned that the paper states 'We show that when training the binary discriminator between in- and out-distribution together with a standard classifier on the in-distribution in a shared fashion, the binary discriminator reaches state-of-the-art OOD detection performance' as a contribution and that this claim is not supported by the results in the paper. The authors say they are happy to drop this particular statement and emphasize that their contribution is that that a binary classifier can be a useful tool for OOD detection. The reviewer is not satisfied by this response, as the reviewer feels that this makes the contribution much less impactful.

- Reviewer iH61 (recommending 6: marginally above, initially reject) pointed out that the significance of one of the contributions is limited, since the claims resemble the ones by Thulasidasan et al. [2021] and Mohseni et al. [2020], and initially recommends to reject. The authors respond that those two papers only aim at good performance, but do not unify existing approaches, as the paper under review does. The reviewer slightly raised their score, but again points out that the previous works already show that a binary discriminator performs well.

- Reviewer Lwwq (recommending 10: strong accept) appreciates the unification of different methods and votes for strong acceptance. The reviewer also points out that he/she is not an expert in the field, and thus this reviewer's rating should be taken with care.

- Reviewer YRfA (recommending 8: accept) points out that the authors make notable progress towards a better understanding of OOD methods, but is concerned about what problem the authors are trying to solve and its significance, and states that he/she cannot judge the importance of the paper.

- Reviewer vYWv (recommending 6: marginally above, initially recommending reject) finds that the paper provides helpful insights to connect methods for OOD detection tasks, and weakly recommends acceptance.

The reviewer's opinions on this paper vary significantly. Initially, a major selling point of the paper was that 'the binary discriminator reaches state-of-the-art OOD detection performance', but after discussion, the authors and reviewers agree that this statement is not supported by experiments, and the idea of using a binary discriminator is also not new, and thus everyone agrees that this statement should be removed.
This leaves as the major contribution an improved understanding of a variety of methods, and casting them as versions of a binary classifier.
This by itself would be sufficient to carry a paper, however the stated equivalence is rather weak as it is based on an asymptotic analysis, and in the asymptotic regime, out-of-distribution detection is rather trivial because the distributions are given. This also explains why in the paper's experiments all the methods that are asymptotically related behave quite differently in experiments.

I do not recommend this paper for acceptance. I've read the paper and I've thought quite a while it and its reviews. I have also discussed the paper with a colleague who works actively on out-of-distribution detection, since I'm not an expert on this topic myself. While in general I find it very valuable to unify and to understand existing out-of-distribution algorithms better, I don't see how the particular interpretation provided by the paper is impactful, since it is unclear how the connection drawn in an asymptotic setup for Bayes classifiers actually extend to concrete OOD detection algorithms, which operate in the finite sample regime.